# Roof renewal disparities widen the equity gap in residential wildfire protection

Sebastian Reining [1] ✉, Moritz Wussow[1,2], Chad Zanocco[2] & Dirk Neumann [1]

Wildfires are having disproportionate impacts on U.S. households. Notably, in California, over half of wildfire-destroyed homes (54%) are in low-income areas. We investigate the relationship between social vulnerability and wildfire community preparedness using building permits from 16 counties in California with 2.9 million buildings (2013–2021) and the U.S. government's designation of disadvantaged communities (DACs), which classifies a census tract as a DAC if it meets a threshold for certain burdens, such as climate, environmental, and socio-economic. Homes located in DACs are 29% more likely to be destroyed by wildfires within 30 years, partly driven by a gap in roof renewals, one of several important home hardening actions. Homes in DACs have 28% fewer roof renewals than non-DACs and post-wildfire, non-DAC homes have more than twice the increase in renewals (+17%) compared to DAC homes (+7%). Our research offers policy insights for narrowing this equity gap in renewals for wildfire-prone areas. We recommend increasing financial support for roof renewals and targeted awareness campaigns for existing programs which are not sufficiently emphasized in wildfire strategies, particularly in DACs.

Changing climate conditions and expanding human development within the Wildland-Urban Interface (WUI) are contributing to increases in wildfire impacts worldwide[1–4]. This is typified by the Western U.S., where communities have faced substantial increases in the intensity of wildfires over the past two decades, leading to considerable economic, ecological, and health impacts[5–8]. Among western states, California has been particularly affected, with 13 of the 20 most destructive fires in its history occurring since 2017[9]. Concerningly, growing evidence suggests that impacts from wildfires are inequitable, with vulnerable communities bearing more burdens from fires[10,11], and also being disproportionately exposed to wildfire smoke[12].

However, addressing this equity gap in wildfire impacts is complex and requires an understanding of both natural processes and human actions. This is particularly challenging because there is evidence that some of the proposed solutions are inequitably distributed across communities. For example, there are disparities in public investment for wildfire risk reduction[13], firefighting resource allocation[14], and household mitigation actions[15,16]. Wildfire preparedness is a multifaceted and

perpetual process that encompasses risk reduction and prevention, as well as preparation for potential evacuations. The most critical components of wildfire risk reduction for homeowners involve vegetation management, the reduction of fuels, and home hardening, which is the utilization of fire-resistant construction materials and design modifications to make structures less susceptible to fire[17,18]. For example, homeowners should use non-combustible siding, fences, and decking materials, install multi-pane windows with non-combustible shutters, and choose fire-resistant roofing[19]. Traditionally, the literature exploring wildfire preparedness has relied on household surveys and case studies[20], which are unable to capture the full geographic and temporal scale of the issue, as household mitigation actions are difficult to observe at scale. To address this observed gap in community impacts, we focus on the adoption of a critical household wildfire protection action: fire-resistant roofing, which can be observed by analyzing building permits. Fire-resistant roofing is identified as one of the most effective home hardening measures against wildfires[18,21,22], yet it is unknown if fire-resistant

[1]Climate Action Research Lab (CARL), Chair of Information Systems Research, University of Freiburg, 79098 Freiburg im Breisgau, Germany. [2]Civil and Environmental Engineering, Stanford University, Stanford, CA 94305, USA. ✉e-mail: sebastian.reining@is.uni-freiburg.de

roofs are distributed equitably and what implications this may have on future wildfire impacts in communities.

Residential preparatory actions are critical for improved community protection because the spread of fire within communities is strongly related to the combustibility conditions of structures and their surrounding environment[23]. During wildfires, structures can be threatened by direct flame contact, radiant heat, and, most importantly, flying embers, which are the primary cause of residential building destruction[24–26]. Structures located within 700 meters, and in some cases even as far as 2000 meters, from the firefront can be destroyed during a wildfire[27,28], and roofs, with their large surface areas, are particularly vulnerable to ignition[29,30]. We find that fire-resistant roofs could reduce the risk of destruction of a residential building if exposed to a nearby wildfire by 8 to 27 percentage points (p.p., 25th to 75th percentile). The level of a community's protection, however, is greater than the sum of its individual measures, as the benefits of wildfire protection measures extend beyond fire-resilient properties to neighboring structures[31]. For instance, our analyses show that the risk of a fire-exposed residential building being destroyed could be reduced by 4 to 8 p.p. if its neighbors have fire-resistant roofs, and during the 1991 Oakland Hills fire, it was found that each burning non-fire-resistant roof led to the ignition of ten additional homes[29,32]. Consequently, California implemented new building code regulations in 1995, mandating the use of fire-resistant materials, and further reinforced these codes in 2008, making them the strictest in the U.S.[33]. For example, according to the Building Code (Chapter 7 A)[34] all re-roofings in high wildfire risk areas in California must use materials with the highest levels of fire resistance and must modify the eaves and vents to minimize the accumulation of flammable materials.

While these regulations reduce the risk of wildfire damages[35], they also increase the costs of home retrofits on average by more than $7000[36,37], presenting a substantial financial investment for homeowners. Moreover, there is less financial support available for home hardening compared to vegetation management, as most of California's grant programs related to wildfire protection focus on the latter[38] (Supplementary Note 1). In addition to costs, other factors associated with the participation in wildfire mitigation actions include building characteristics, the biophysical environment, perceived risk, actual risk exposure and occupants' sociodemographic characteristics[15,39–42] (Supplementary Note 2 for an extended literature review). High wildfire risk areas are typically considered less socially vulnerable[43–45]. Nonetheless, recent research also indicates that communities facing higher levels of past fire exposure often have lower income levels[10], as socially and economically disadvantaged communities may have less capacity to take protective actions[46]. Additionally, socially vulnerable communities may also bear other impacts from wildfires, such as lower future earnings due to smoke-related reductions in educational outcomes[47]. This underscores that vulnerability is a compounding process, where limited resources for recovery can increase a community's susceptibility over time[48].

In this study, we explore five critical questions: (RQ1) Are there disparities in wildfire-induced property damages? (RQ2) How effective are fire-resistant roofs in reducing wildfire-induced property damage? (RQ3) How equitable is the level of investment in roof renewals? (RQ4) What is the effect of nearby wildfire exposure on roof renewals? (RQ5) What are the potential future equity impacts of roof renewals?

To answer these questions, we investigate the relationship between the adoption of fire-resistant roofing, wildfire exposure, and social vulnerability by applying the disadvantaged communities (DACs) designation from the U.S. government's Justice40 initiative[49]. A census tract is designated as disadvantaged if it meets the threshold for at least one category of environmental or climate-related burdens – such as climate change, energy, health, housing, legacy pollution, transportation, water and wastewater, or workforce development – and also falls below a socioeconomic threshold in terms of income or education. Additionally, communities within the boundaries of Federally Recognized Tribes or tracts surrounded by disadvantaged communities that meet an adjusted low-income threshold are also considered disadvantaged. Our study focuses on California, due to the availability of comprehensive data, strict building codes, and since the state accounts for around 70% of the U.S.'s wildfire-induced housing destruction (Supplementary Note 3).

Here, we find substantial disparities in historic wildfire exposure in California, with more than 54% of the residential structures destroyed by wildfire between 2013 and 2021 located in communities with a median income below the 30th percentile. Based on the analysis of building permits from 16 counties, the rate of roof renewals per building is around 28% lower in DACs compared to non-disadvantaged communities (non-DACs). In addition, we observe similar equity gaps in other preventive approaches such as the Firewise Community program, which aims at fostering local, homeowner-driven measures. Our research also indicates statistically significant differences in homeowner responses to wildfire exposure in terms of roof renewals. Within three years of a wildfire, non-DACs experience a 17% rise in roofing permits, whereas DACs have less than half this increase (+7%). Furthermore, we show that DACs face a much higher risk of wildfire-related destruction on a building level over the next 30 years (+25%). The discrepancy in destruction risk could be further expanded if we account for differences in roof renewal rates (+29%). Finally, we explore how increases in the roof renewal rate of DACs can reduce this equity gap. Our results suggest that by focusing on the most at-risk properties, elevating DACs' roof renewal rates from the projected 22% to over 50% in the next 30 years − surpassing those in non-DACs − could close the gap in expected destruction rates for disadvantaged communities.

## Results
### Equity gap in wildfire-related destruction of residential buildings
We find that wildfires disproportionately impacted lower-income communities in California, as shown by the proportion of wildfire-destroyed buildings in DACs compared to non-DACs from 2013 to 2021 (Fig. 1A). Communities in the lowest three income deciles sustained over 54% of the total residential building destruction (see Supplementary Note 17 for robustness tests), whereas communities in the highest three deciles incurred 27%, based on our analysis of the CAL FIRE DINS database. This disparity is accentuated at a per-property level due to the greater number of buildings in higher-income communities[50,51].

In light of these impacts, we analyzed the association between fire-resistant roofing and wildfire-induced destruction of residential buildings (Fig. 1B). We find that the average fire-resistance score of wildfire-affected buildings is negatively correlated with destruction, more so in DACs ($-0.63$, $p = 0.0013$) compared to non-DACs ($-0.45$, $p = 0.012$). We further model the impact of roof characteristics on the probability of a residential building getting damaged when exposed to a wildfire (Fig. 1C). Our results indicate that buildings with the most fire-resistant roofs have a reduced risk of destruction ranging from 8 to 27 p.p. (25th to 75th percentile), with an average reduction of 18 p.p. (Fig. 1C, Supplementary Note 5). This corresponds to a median reduction of 26%, which is consistent with prior research[24,52,53], estimating effects in the 12–33% range. Due to the stringent roofing regulations for fire-prone areas in California[35], newly installed roofs are likely to provide risk reductions at the upper end of this spectrum. Moreover, the risk of a building being destroyed can be reduced by 4 to 8 p.p. (mean: 6 p.p.) if its neighbors have fire-resistant roofs. We control for other construction features, building age and structure types such as mobile homes, as well as neighborhood characteristics. Building age is strongly associated with damage probability, with newer buildings, on average, being less likely to get damaged (Supplementary Notes 5, 13). This finding is consistent with the protective

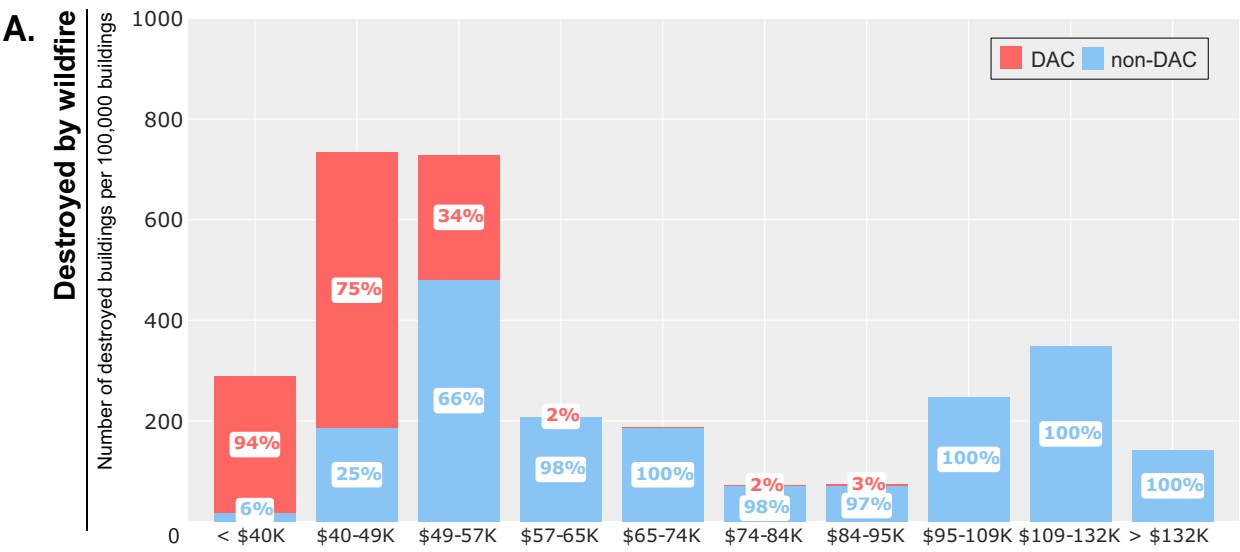

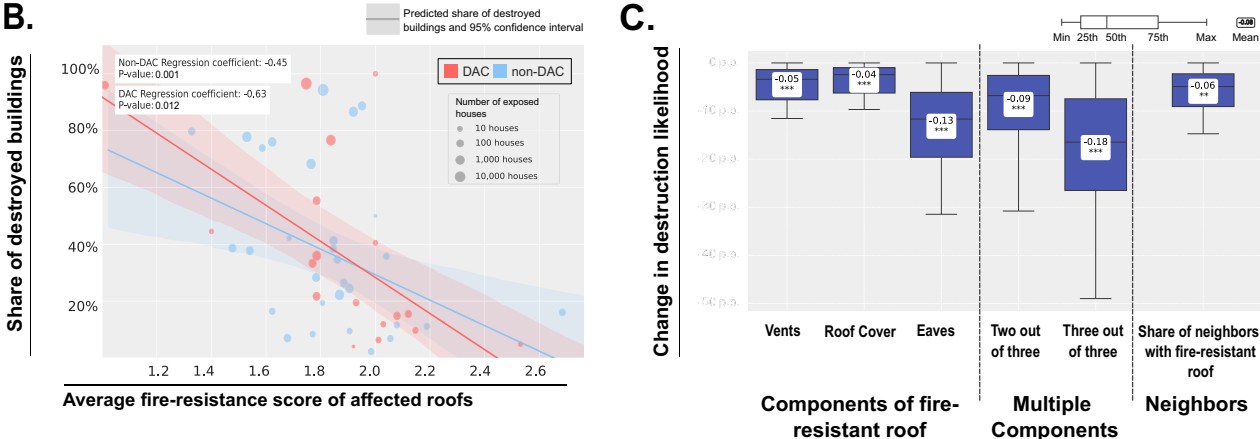

**Fig. 1 | Residential buildings destroyed by wildfire and risk reduction of fire resistant roofs. A** Residential buildings destroyed by wildfire in California (2013-2021) by median income groups. Includes all residential buildings that are considered 'destroyed' based on CAL FIRE's DINS database. Buildings are spatially merged with 2010 census tracts and income deciles are calculated using the population-weighted median household income based on the 2015–2019 American Community Survey[73]. Classification into disadvantaged communities (DAC) and non-disadvantaged communities (non-DAC) is based on the definition of the U.S. government's Justice40 initiative[49], as implemented in its Climate and Economic Justice Screening Tool (CEJST). The sample includes $n = 33,689$ residential buildings. Measure of income deciles and destroyed buildings per 100,000 buildings includes 8056 California census tracts (Methods). **B** Share of destroyed residential buildings per wildfire event and the average fire-resistance score of the affected buildings by DAC status. Each point corresponds to a wildfire event, the average fire-resistance score of all exposed residential buildings (undamaged and damaged) is displayed along the x-axis and the share of those buildings that were

destroyed during the wildfire event is displayed on the y-axis. The fire-resistance score is measured on a scale of 0 to 3 and it is based on the number of fulfilled criteria among the following: roof cover (is either asphalt, metal, concrete, or tiles), vents (mesh screens > 1/8) and eaves (no eaves or enclosed eaves). The sample includes $n = 40,673$ residential buildings exposed to wildfire. **C** Effect of fire-resistant roofs on wildfire-induced destruction likelihood of residential buildings. Displayed values are the predicted percentage point changes based on the results of a conditional logit regression model that quantifies the association between fire-resistant roofs and the probability of a building being damaged by a wildfire conditional on wildfire exposure. Negative values signify a decrease in destruction likelihood. The model includes controls for further building characteristics, construction year, structure type (e.g. single-family vs. mobile home), and fire event fixed effects (Methods and Supplementary Note 5). The sample includes $n = 40,673$ residential buildings exposed to wildfire. Statistical significance levels are *$p < 0.05$, **$p < 0.01$, and ***$p < 0.001$.

effect of Californian building codes, which were strengthened again in 2008 and require the implementation of several fire-resistant measures, including roofs[35].

### Disparities in roof renewals for disadvantaged communities (DACs)

Many of the low-income and fire-impacted communities have marginalized or underserved populations, or are overburdened by

pollution, and have been recognized as disadvantaged by the U.S. government. Disadvantaged communities may also have less capacity for investing in wildfire preparedness, such as effective home hardening measures like roof renewals[54]. We find that across time DACs consistently lag behind non-DACs in terms of roof renewals, with the disparity widening during the COVID-impacted years 2020 and 2021 (Fig. 2A). In these years, DACs had approximately half the renewals of non-DACs per census tract (−44%), and after accounting for the

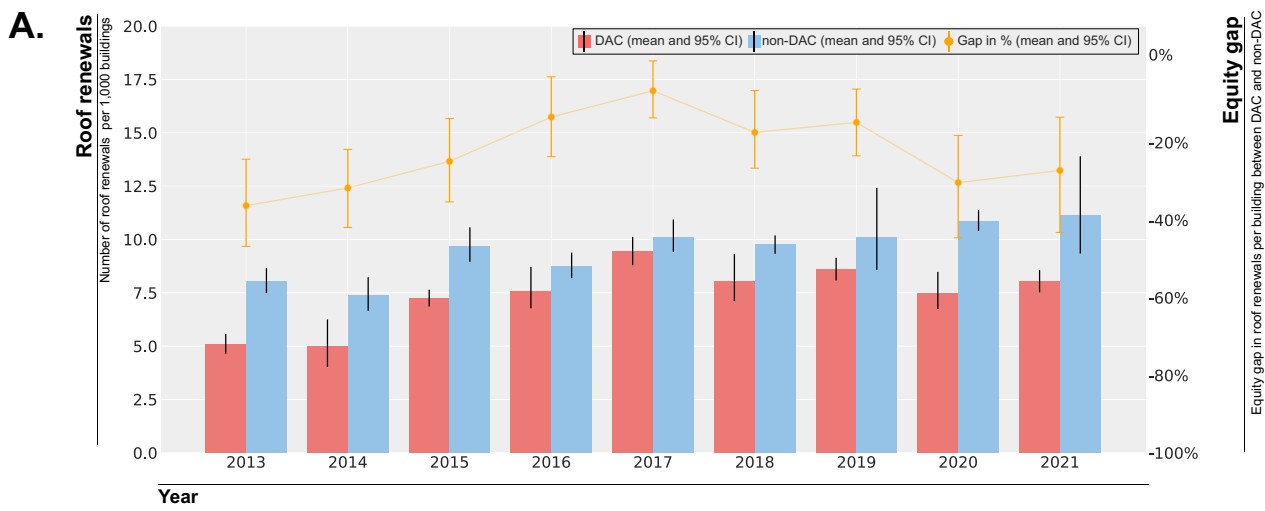

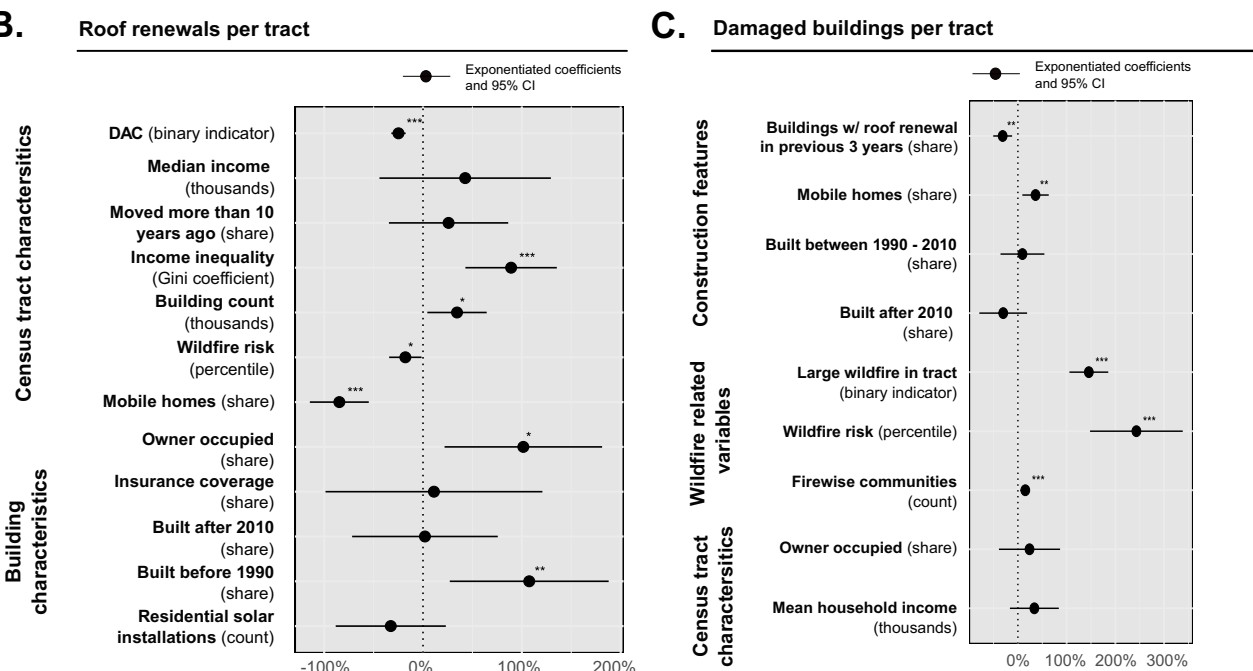

**Fig. 2 | Roof renewals per year in California and regression results for the number of roof renewals within a census tract. A** The mean number of roof renewals per 1000 buildings per year by disadvantaged community status (bar chart) with black error bars representing 95% confidence intervals estimated via bootstrapping. Gap between disadvantaged communities (DAC) and non-disadvantaged communities (non-DAC) is calculated as the percentage difference between the means (orange line chart) with orange error bars representing 95% confidence intervals estimated via bootstrapping. The sample includes $n = 2563$ census tracts from California. **B** Exponentiated coefficients from a negative binomial count regression representing the percentage change in the expected number of roof renewals per census tract for a one-unit change in the independent variables. In addition to the displayed independent variables, the model includes spatial and temporal fixed effects at the county and year level. The sample includes $n = 2563$ census tracts from California that were observed over a 9 year period. All

independent variables except for the building count are normalized to the [0,1] interval so the coefficients represent the difference in effect size when the corresponding variable is at its maximum (coded as 1) vs. minimum (coded as 0). **C** Exponentiated coefficients from a negative binomial count regression representing the percentage change in the expected number of buildings with wildfire-induced damages per census tract for a one-unit change in the independent variables. In addition to the displayed independent variables, the model includes spatial and temporal fixed effects at the county and year level. The sample includes $n = 751$ census tracts between 2015 and 2021 from California that were exposed to wildfires larger than 300 acres. All independent variables are normalized to the [0,1] interval so the coefficients represent the difference in effect size when the corresponding variable is at its maximum (coded as 1) vs. minimum (coded as 0). Statistical significance levels are $*p < 0.05$, $**p < 0.01$, and $***p < 0.001$.

number of existing buildings, still 28% fewer roof renewals. The disparities between DACs and non-DACs remain statistically significant in a panel regression including county and year fixed effects and controlling for socio-demographic and building-related factors such as the number of mobile homes (Fig. 2B). DAC status is associated with a 32% ($p < 0.001$) lower roof renewal rate compared to non-DACs (see Supplementary Note 4 for alternative model specifications). Fire risk is

negatively correlated with the number of roof renewals, potentially related to stricter regulations that require more expensive roof installations. In contrast, areas with higher income, more new building construction, and a higher share of owner-occupancy, are associated with more roof renewals. This aligns with survey-based literature suggesting that financial capacity and homeowner presence are important drivers of wildfire protective actions[55], suggesting that a

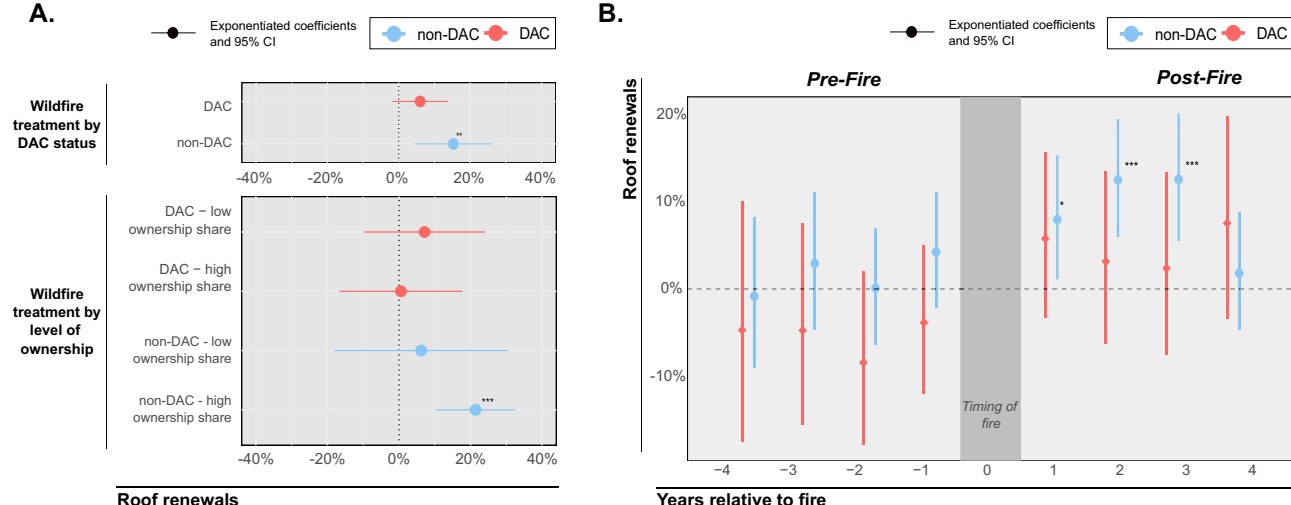

**Fig. 3 | Heterogeneous treatment effects of wildfire exposure on roof renewals between DAC and non-DAC. A** Exponentiated coefficients and 95% confidence intervals of a negative binomial regression in a difference-in-differences setting of wildfire treatment on the number of roof renewals within a census tract. A tract is considered treated if the population-weighted distance to a wildfire was below 10 km in the last three years. The binary treatment variables are then interacted with DAC and non-DAC status and the level of ownership of occupied buildings within a census tract. We classify a tract as having a low ownership share if the share of owner-occupied housing units is below the 25th percentile and as high ownership if it is above the 75th percentile. **B** Exponentiated coefficients of a negative

binomial regression for the number of roof renewals relative to a fire in DACs vs. non-DACs. The exponentiated coefficients displayed in this plot represent the percentage change in the number of roof renewals per census tract for a one-unit change in the independent variables. Statistically insignificant effects prior to a fire confirm identification via the difference-in-difference setting for our main specification. The model includes control variables for building characteristics, socio-demographics, and fixed effects at the census tract and year level. The sample includes $n = 2523$ census tracts across 9 years. Statistical significance levels are $*p < 0.05$, $**p < 0.01$, and $***p < 0.001$.

care factor – the willingness and ability of homeowners to invest in protective measures – may play a role in the decision to renew a roof. Our findings are also robust to the use of different measures of disadvantaged communities (Supplementary Note 14). Our dataset includes roof renewal information, building features, and socio-demographic information for over 2.9 million residential buildings for the 2013 to 2021 period. It was obtained by filtering and geo-locating building permit data from 16 California counties and spatially merging them to census tracts.

We next assessed the relationship between roof renewals and wildfire-induced damages and our analysis shows a statistically significant, negative effect. Therefore, we evaluated the association between the share of buildings with roofs within a census tract that were renewed within the previous 3 years and the number of buildings that were damaged by wildfires (Fig. 2C). In the event of a wildfire, the number of wildfire-damaged buildings is 19% lower when the share of roof renewals is at its maximum compared to its minimum (p < 0.01). Our analysis focuses on the census tracts that were exposed to wildfires between 2013 and 2021, covering approximately half of the overall damage to residential buildings caused by wildfires in California during this period.

### Effects of wildfire exposure on roof renewals
We find a statistically significant treatment effect of wildfire exposure on roof renewals. A census tract is classified as treated when its population-weighted distance to a wildfire has been less than 10 kilometers at any point in the past three years. Wildfire exposure increases the number of roofing permits in a tract in the following three years on average by 12% (95% CI: 8% to 16%, p < 0.001) (see Supplementary Note 4 for detailed regression tables). However, we find substantial differences in the treatment effect between DAC and non-DAC tracts. Post-wildfire, non-DACs experience a 17% increase in permits (p = 0.008, Fig. 3A), while roof renewals within DACs increase by only 7% (not statistically significant, p = 0.13). Splitting communities by the share of owner-occupied households, we find no statistically

significant interaction effect across all DACs as well as for non-DACs with a low share of owner-occupied households. However, for non-DACs with a high share of owner occupancy, the number of new roofs increases by 22% post-wildfire exposure (p < 0.001).

We further employ a negative binomial panel regression with census tract and year fixed effects and find a statistically significant increase in non-DAC roof renewals for each of the first 3 years following a wildfire treatment (Fig. 3B). These results are consistent across alternative model specifications, including county-year fixed effects, different treatment lengths and distance cutoffs, alternative ownership limits, as well as other exposure definitions, such as the number of buildings exposed to a wildfire. In particular, our analysis is not confounded by post-fire rebuilding efforts as we have excluded permits near damaged properties, and conducted placebo tests for pre-wildfire treatments that show no statistically significant effects (Supplementary Note 4). This design is essentially a difference-in-difference set-up with staggered treatments, which is motivated by a common trend of treated and untreated tracts prior to a fire.

### The potential of roof renewals for reducing wildfire-related building destruction in DACs
To assess the implications of the roof renewal equity gap, we simulated expected wildfire damage in California over the next 30 years. Our analysis, which is based on historic building permits, indicates that 22% of buildings in DACs and 30% of buildings in non-DACs are expected to get roof replacements over a 30-year-period (Supplementary Fig. 8). Utilizing First Street Foundation's wildfire risk model[56], we project future wildfire exposure for each census tract and estimate the average proportion of homes destroyed by wildfires. We find a consistent disparity in wildfire destruction among communities, particularly in the most wildfire-prone areas. With rising wildfire risk per tract, the expected destruction of housing increases from below 0.5% for low and medium-risk areas to above 2.5% for very high-risk areas (Fig. 4A). Although non-DAC tracts see a greater absolute number of buildings destroyed (Supplementary Note 6), DACs have a 25% higher

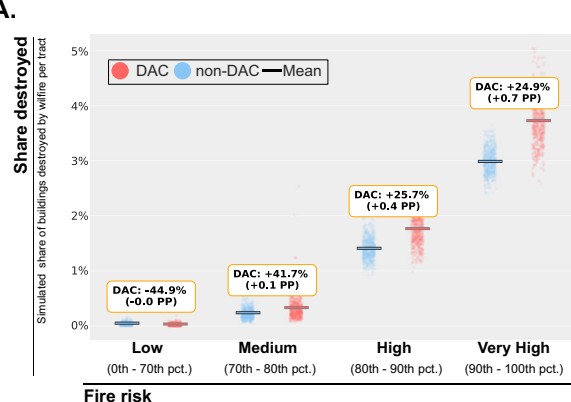

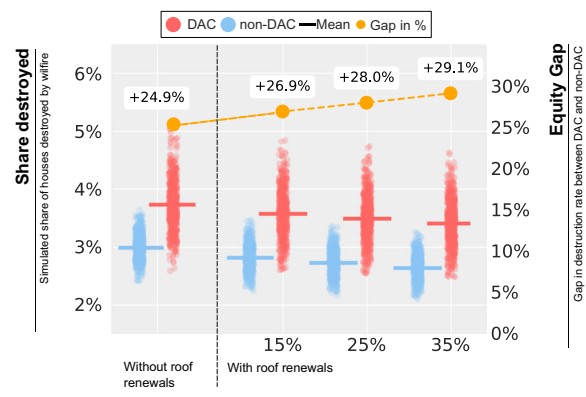

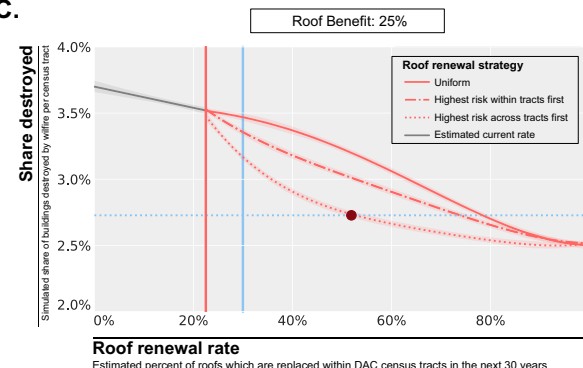

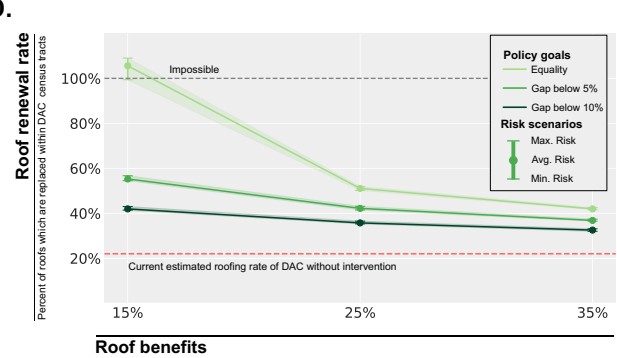

**Fig. 4 | Counterfactual analysis: Impact of roof renewal rates on disparities in wildfire-related destruction of buildings in California. A** Simulation of the expected share of destroyed buildings per census tract over the next 30 years split by wildfire risk and DAC status. The Justice40 fire risk percentile (pct.) categories and 30-year wildfire exposure risks are based on First Street Foundation's wildfire model. The figure depicts the share of destroyed buildings per simulation run (points) and the mean across 1000 runs (vertical lines). The gap is calculated as the percentage difference between the means. The sample includes $n = 8046$ census tracts from California. **B** Simulating the impact of the different roof renewal rates on the equity gap in terms of the share of destroyed buildings for DACs vs. non-DACs within the highest wildfire risk category ("Very High" fire risk in Subfigure A). Roofs are assumed to reduce the probability of destruction given wildfire exposure by 0–35%. The sample includes $n = 1991$ census tracts with the highest fire risk category from California. **C** Simulation of the average share of destroyed buildings across roof renewal rates within DACs with the highest fire risk in California ("Very High" fire risk in Subfigure **A**). The destroyed share of buildings declines as more roofs are renewed. Roof renewal strategies differ by the order in which buildings are selected for roof renewals. Roof renewal is assumed to yield a reduction in destruction risk by 25%. The sample includes $n = 1991$ census tracts with the highest fire risk category from California. **D** Required roof renewal rate to reduce gap to 10%, 5% or 0% across different risk scenarios and roof renewal benefits. The required roof renewal rate is calculated as the intersection of the simulated DAC share of destruction with the corresponding non-DAC share of destruction (Methods). The sample includes $n = 1991$ census tracts with the highest fire risk category from California.

destruction rate per property (0.7 p.p.) in high-risk zones. If we factor in the protective benefits of new roofs, the overall rate of destruction could be reduced by up to 8% (0.3 p.p.) for DACs, however, the gap between DACs and non-DACs could widen to as much as 29% due to the differences in roof renewal rates (Fig. 4B). While these results are based on the average projected wildfire risk over the next 30 years, this gap remains in the 25% to 30% range if we simulate based on lower or higher risk scenarios (Supplementary Fig. 9).

To investigate the potential benefits of higher roof renewal rates within DACs, we simulated the expected share of destruction per census tract as a function of the roof renewal rate (Fig. 4C). We considered variations in the estimated benefits of renewed roofs, risk scenarios, and strategies for roof replacement, which determine the prioritization of renewals. Consistent with our assessment of roof renewal benefits and findings of prior research[35,57], our model also includes spillover benefits for non-upgraded homes (Supplementary Note 11). Our analysis uses a dataset that classifies all buildings within a census tract into ten fire exposure categories, enabling us to simulate and assess the impact of prioritizing roof renewals across these categories using different strategies. The first strategy uniformly distributes new roofs across all risk levels within a census tract

("uniform"), leading to a steady, albeit incremental, decrease in destruction. The second strategy, "highest risk within tract", focuses on the most vulnerable homes within each census tract, yielding a more pronounced reduction in risk. The most effective approach allocates renewals to the most endangered homes across all tracts ("highest risk across tracts"), highlighting the compounding benefits of improved homes on neighboring homes.

Figure 4C illustrates three core findings: First, a higher number of roof renewals could substantially reduce the expected share of destroyed buildings over a 30-year period (from 3.8% to as low as 2.5% in case of a 100% roof renewal rate). Second, the reduction in risk generated by each additional new roof is dependent on both the roof distribution strategy and the existing rate of renewal. Targeted approaches (e.g., "highest risk across tracts") exhibit a convex decline in marginal benefit, in contrast to the rather concave shape seen with a uniform renewal distribution. Finally, to bridge the gap in destruction rates between DACs and non-DACs, the rate of roof renewals in DACs must exceed that in non-DACs. Even with conservative projections of roof renewal benefits, a substantial mitigation of wildfire risk could be achievable and the equity gap could be effectively narrowed (Fig. 4D). This reduction, however, is subject to diminishing returns as the

marginal utility of additional roof renewals decreases. In scenarios where the protective benefits of new roofs are small, roof renewals alone are insufficient to bridge the equity divide, necessitating additional interventions. Still, the gap could be reduced considerably to as low as 10% or 5%.

## Discussion

Our study reveals a substantial equity gap in wildfire-related destruction of residential buildings in California, with lower-income areas suffering the most structural damage, which is consistent with prior findings on communities affected by fires[10] (RQ1). These communities' vulnerability is often exacerbated by a lack of resources for effective post-wildfire recovery and rehabilitation[48]. The discrepancy in wildfire damages may stem from various factors such as differences in public investment in risk reduction projects[13], unequal firefighting priorities[14], and the impact of repeated wildfires on income[10]. However, reverse causality has also been observed, where socio-economic factors contribute to a rise in wildfire incidents due to rural abandonment and arson[58]. Our research offers another explanation, as we document an equity gap with respect to investments in home hardening, particularly by analyzing roof renewals. These are important indicators of home hardening in wildfire-prone areas and we find that new roofs could reduce the likelihood that a residential building gets destroyed when exposed to a wildfire by up to 26% (RQ2). On average DACs have 28% fewer roof renewals on a per-property level than non-DACs (RQ3). Given the high replacement costs for a roof[36,37], this aligns with earlier survey-based research indicating that costs are the primary barrier to implementing structural measures[55]. Our findings indicate that exposure to nearby wildfires motivates non-DAC homeowners to invest in roof renewals (RQ 4): within the 3 years that follow an exposure to a wildfire and within a radius of 10 kilometers, roof renewals increase by 22% on average in non-DACs with high home ownership rates. While non-DAC tracts with low home ownership rates and DACs also show increased re-roofing activity after wildfire exposure, the observed effects are substantially smaller and not significant at the common $p < 0.05$ statistical significance threshold.

The differences between DACs and non-DACs also extend to less-costly measures, such as forming Firewise communities, which are likely to enhance vegetation management. Consistent with previous literature[15], we document that DACs have disproportionately fewer Firewise communities (Supplementary Note 7). The voluntary program by the NFPA promotes local safety solutions by encouraging homeowners to jointly reduce wildfire risks. Within the areas with the highest risks, 35% of tracts are disadvantaged, but only 16% of Firewise communities are situated there, a gap that may further exacerbate differences in the capacity to mitigate wildfire risk between DACs and non-DACs. Homeowner associations play a key role in setting up Firewise communities, yet these associations are less prevalent in low-income areas[59]. Taken together, this suggests a gap in both information provision and capacity for fire mitigation in these regions.

The equity gap in wildfire preparedness is particularly concerning given the increasing wildfire risks throughout the U.S. and in other parts of the world. Our simulations of building destruction over the next 30 years reveal a pronounced disparity between DACs and non-DACs (+25%), further amplified by the variance in roof renewals (up to +29%, RQ5). Yet, our analysis also demonstrates that strategically directed public investments in home hardening can substantially mitigate structural losses. To reach equity in terms of destruction in the high wildfire risk zones of California, our base scenario shows that upgrading around 527,000 buildings (52% roof renewal rate) in DACs would be necessary. Assuming an average cost of $20,000 per roof and potential funding mechanisms covering 30%, the financial investment need stands at $105 million annually – if the effort is distributed evenly across the next 30 years. These investments could save approximately 10,000 buildings in DACs, valued at $5 billion,

representing a net gain of $2 billion for the economy (Supplementary Note 12). While the projected costs are considerable, they appear modest when viewed in the broader fiscal context – California's CAL FIRE annual base budget alone is $2.9 billion, with peak years incurring over $1 billion in supplemental wildfire suppression costs[60]. Also, California already allocates hundreds of million dollars annually to wildfire prevention grants, which primarily target vegetation management[38] (Supplementary Note 1). Existing local initiatives offer only limited home hardening funds, typically between $2000 and $5000, far short of the average renewal cost for a roof.

Investing in home hardening and especially new roofs creates other important benefits. California's insurance market is characterized by soaring premiums and the retreat of insurers from high-risk zones[61], and roof renewals could help with stabilizing insurance rates in these areas. Additionally, while the relative benefits of roofing may seem less immediate compared to other preventive measures such as vegetation management, roofs offer a long-lasting impact with less ongoing maintenance. Furthermore, the adoption of modern roofing technologies increases home energy efficiency, and recent research suggests energy savings of up to 32%[62]. Relatedly, a roof renewal can improve a household's readiness for rooftop solar, contributing to broader energy cost savings and the achievement of sustainability goals[63].

Our findings yield practical insights for policymakers. First, focusing resources on high-risk communities reduces risk more effectively than a widespread distribution, as the risk reduction benefits of a new roof extend to neighboring properties, thereby enhancing the collective impact[31,57]. Second, our findings highlight the experiential learning of homeowners, as roof renewals increase on average by 12% in the three years following wildfire exposure. The observed treatment effect is in line with the Protection Motivation Theory[64] (Supplementary Note 2 for extended literature review). The theory posits that as individuals update their assessments of risks, their motivation to invest in protection (i.e., new roofs) increases. Thus a "window of opportunity"[65] for public investment arises in the aftermath of a wildfire, during which homeowners are particularly inclined towards preventative actions. Our study substantiates the observed behavioral shifts post-disaster with empirical evidence, adding to the body of work on disaster response behaviors[64,66]. Third, our research suggests directing more funds towards home hardening in disadvantaged communities. Interestingly, DACs are not inherently averse to investing in fire prevention; our findings indicate that once established, Firewise communities in DACs are similar to those in non-DACs in terms of the number of members and the probability of staying active, and while average investments are lower in DACs such disparities decline with increasing exposure to fire risk (Supplementary Note 7). We therefore argue that DACs need the support of targeted policies to help them take the first steps toward enhanced wildfire preparedness[38,46].

Our study is not without limitations, offering avenues for future research to explore. First, while our focus on roofing and Firewise communities addresses key elements of wildfire preparedness, it is by no means exhaustive; other home hardening measures such as wall, window, and door upgrades[17] as well as vegetation management are also important for effective wildfire resilience. Homeowners and policymakers should note that comprehensive fire vulnerability assessments of structures are necessary, and roofs should be considered just one element among many actions. Roof renewals alone do not guarantee home protection. Second, the large variation in the frequency and severity of wildfires[67] can lead to a small number of highly destructive events disproportionately affecting the results. Still, our main results are robust to the exclusion of the largest fires (Supplementary Note 5 and 15). Third, our analysis relies on building permit data, which, due to variation in data availability and quality, likely underestimates the true roof renewal rate. Around 20% of permits

could not be located and hence were excluded from our analysis. This likely only affects the magnitude of the renewal rates and does not introduce bias, as our regression analyses incorporate spatial fixed effects at the county or census tract level, helping to mitigate any discrepancies in how permits are processed across different counties. Additionally, our dataset lacks granular information on the installed base and quality of homes, as we only observe some building features, such as construction year, at the census tract level and even for older buildings of the same construction age there can be varying levels of fire resistant roofing materials (e.g., clay tile). Lastly, our counterfactual simulation should be seen as a data-driven evaluation to assist policy-making rather than an exact projection of future wildfire impacts. It does not account for new construction, which tends to be higher in non-DACs and typically features homes of better quality and fire resilience[50]. Our model also simplifies the estimation of home destruction by clustering at the census tract level instead of assessing wildfire destruction risk and roof benefits on a building-by-building basis. This presents an opportunity for more nuanced analysis using detailed building features that could potentially be derived from aerial and street view imagery[68]. Parcel-level vulnerability analysis could determine which wildfire exposure types are most likely and which home hardening measures have the greatest impact.

While our geographic focus is on California, we believe that our research offers valuable insights beyond the state's borders. Historically, California has been the state most prone to destructive wildfires, as climate change leads to drier, warmer conditions, areas less accustomed to wildfires are increasingly at risk[69]. The devastating wildfires in Hawaii in 2023, destroying over 2,200 structures and causing over a hundred fatalities[70], and the 2024 California Park Fire, which to date is among the largest in the state's history[71], highlight this escalating threat. Areas in the U.S. and beyond that face future wildfire risk should therefore act preemptively and adapt building codes as well as offer necessary financial support for homeowners. Proactive policy action is crucial, as homeowners increasingly shift responsibility to governments[55], and a change in building codes might take decades to yield a widespread presence of hardened homes. Future work could quantify the collective benefits of group versus single-home roof renewals and delve into how different types of resilience measures can be integrated most effectively to support fire-resistant communities.

Our research argues for a substantial expansion of new and existing home-hardening policy initiatives in wildfire-prone communities, which appear to be currently underemphasized in wildfire defense strategies – particularly so in disadvantaged communities. Additionally, our counterfactual analysis indicates that roof renewals on their own may not suffice to bridge the equity gap in wildfire resilience so it is essential to maintain fire safety as a composite strategy, integrating various protective measures and balancing the allocation of resources. In this respect, the recently launched "California Wildfire Mitigation Program - Home Hardening Initiative", which is funded by CAL OES, CAL FIRE, and local communities shows a promising approach[26]. Currently in its pilot phase, this program aims to provide assistance to disadvantaged homeowners in high-risk wildfire zones in San Diego. The focus is on home hardening and creating defensible spaces, supporting homeowners with up to $40,000. Our research demonstrates the critical nature of such initiatives, highlighting the necessity for policymakers in all wildfire-prone regions to acknowledge and engage with household-level proactive preparedness investments.

## Methods
### Data sources
All data applied in our study were either acquired from public sources or were obtained via Freedom of Information Act and public records requests. For our main analysis, we used seven different data sources. (1) The CAL FIRE Damage Inspection Database (DINS)[72] provides detailed information on over 90,000 structures affected by wildfires in California from 2013 to 2022. It includes information about properties such as their location, extent of damage, and building characteristics including roofing materials. (2) Socioeconomic information was obtained from the American Community Survey's (ACS) 5-year estimates[73]. A census tract is a small, relatively stable geographical unit defined by the Census Bureau, containing between 1200 and 8000 residents and on average around 1500 residential buildings. Tracts are designed to facilitate statistical comparisons and analyses of population and socio-economic data within a region. (3) Classification of communities into disadvantaged (DAC) and non-disadvantaged (non-DAC) and assignment of fire risk groups is based upon the U.S. Council on Environmental Quality's Climate and Environmental Justice dataset[49]. (4) The number of roof renewals was extracted from building permits that are issued by city and county authorities. Assessor parcel maps were utilized to geocode the permits and assign them to census tracts. We gathered a more extensive collection of building permits than those available online by directly contacting the relevant county offices but did not include some databases due to incomplete information (three counties) or non-digitized permits (one county). We focused on residential roofing renewals, excluding solar installations and any repeat permits within the same year. We also excluded 304 census tracts due to concerns of incompleteness, since less than five permits were reported for these tracts. Our final dataset comprises permits for buildings across 16 counties, covering 2563 census tracts for 9 years from 2013 to 2021 (see Supplementary Note 8 and 9 for details). (5) Fire perimeters were extracted using CAL FIRE's fire repository[74]. (6) To simulate the wildfire exposure of census tracts over the next 30 years, we employed publicly available wildfire risk estimates from First Street Foundation[75]. (7) We utilized the location and year of establishment of Firewise communities in the U.S. as identified by the National Fire Protection Association (NFPA)[76]. (8) To approximate insurance coverage within a tract, we used the Community Service Statement from California's Insurance Commissioner and matched their ZIP code level data to census tracts[77]. (9) Information on residential solar adoption was obtained from Berkeley Lab's "Tracking the Sun" (TTS) project[78]. (10) Finally, we used the CalEnviroScreen 4.0 dataset for robustness checks for the measure of disadvantaged communities[79].

### Analysis of wildfire-induced building destruction in California
To quantify the proportion of Californian buildings that were destroyed by wildfire, we analyzed data from CAL FIRE's Damage Inspection (DINS) database[72]. We chose California as a representative state because it accounts for over two-thirds of total wildfire-induced building destruction in the U.S. during our study period (Supplementary Note 3) and the DINS database contains more detailed information than similar databases from most other states. We followed the definition of CAL FIRE and only included buildings that are labeled 'destroyed' (i.e., with a damage assessment of at least 50%), which account for 92% of all damaged residential buildings. We further focused our analysis on residential buildings, thus excluding around 20,000 non-residential destroyed buildings. We classified buildings as residential if their structure type in the DINS database was listed as family homes or residential buildings, including both single- and multi-story structures. Notably, this did not substantially alter the main results of our analysis. Our final dataset of 33,689 destroyed residential buildings was then spatially joined with 2010 U.S. census tract boundaries[80]. Income deciles were computed using population-weighted median household incomes from the 2015 to 2019 American Community Survey (ACS)[73], to align with the boundaries of disadvantaged communities as outlined by the U.S. Justice40 initiative[49]. To account for the margin of error in the median household income estimates provided by the ACS, we used a bootstrapping approach to test the robustness of our main result (Supplementary Note 17). We

then calculated the number of destroyed structures per decile and weighted it by the number of existing buildings within each decile as

$$\begin{aligned} &Weighted\ WildfireDestruction_{Decile} \\ &= \frac{Number\ of\ Destroyed\ ResidentialBuldings_{Decile}}{Number\ of\ ResidentialBuldings_{Decile}} \end{aligned} \quad (1)$$

## Wildfire protection benefits of a new roof

To estimate the wildfire protection benefits that are provided by a new roof, we combined insights from prior research with our own calculations. Our review of the literature suggests that a new roof can reduce the likelihood of wildfire-induced destruction by 12% to 33%[24,52,53] (Supplementary Note 2 for expanded Literature Review), depending on location, building age, and on how a roof is defined (e.g., whether it includes eaves and vents). Our own calculations are based on the CAL FIRE DINS Database, containing information for more than 90,000 structures exposed to a wildfire in California from 2013 to 2022. While for some fires, the database only describes structures reported as damaged, for other fires there is also information on non-destroyed homes. To ensure that per fire there are suitable counter-factuals, we restricted our analysis to fire events where there are at least 100 undamaged buildings recorded in the database. Our final dataset consists of 40,673 residential buildings exposed to wildfire. To create a binary measure, we coded all damaged buildings with a damage value of 26% or higher as 1 (damaged) and otherwise 0 (not damaged). We then ran a conditional logit regression with fire-event fixed effects of various housing characteristics on the assessed damage. Our definition of a fire-resistant roof encompasses not only the roofing material but also eaves and vents, aligning with previous literature[37]. We then estimated the wildfire protection benefits by predicting the change in destruction likelihood for each building when the roof-related variable is set to 0 versus 1. To reflect previous research findings and the uncertainty around the true wildfire protection benefit provided by a new roof, we vary the parameter of reduction in destruction likelihood in our counterfactual simulations between 15% and 35%.

## Relationship of roof renewal rates and wildfire induced damage

To estimate the impact of roof renewal rates on wildfire risk reduction, we applied negative binomial regression to model the logged number of wildfire-damaged residential buildings within a census tract in a given year. The primary independent variable is the share of buildings within a census tract that had a roof renewal within the previous three years.

$$\begin{aligned} Log(Wildfire\ Damages_{Tract,Year}) = f(&\beta Roofs_{Tract,Year} + \gamma X_{Tract,Year} \\ &+ \delta_{County} + \phi_{Year} + \epsilon) \end{aligned}$$
$$(2)$$

Controls include the wildfire risk score as given by First Street Foundation's wildfire model[75], the number of Firewise communities, and whether a tract was exposed to a wildfire in a given year. Additional controls are average building age, the number of mobile homes, and the share of owner-occupied buildings.

Our analysis focuses on census tracts with roof renewal data, which account for 19,673 destroyed or wildfire-damaged buildings between 2013 and 2021, representing 53% of all wildfire damage to residential buildings in California. We restrict our analysis to census tracts exposed to wildfires larger than 300 acres. Our results are also robust to limiting our analysis to census tracts that sustained building damages from wildfires. Supplementary Note 14 provides additional tests with various regression specifications, consistently showing a negative coefficient for roof renewal rates, though not always statistically significant.

## Identifying the association of disadvantaged community status with the number of roof renewal decisions

Due to the regulations in California that prescribe fire-resistant roof renewals, an investment in a new roof represents an investment in wildfire safety. To assess the association of disadvantaged community status with roof renewal decisions, we applied count regression analysis to a panel dataset of roof building permits from 2563 California census tracts that were observed across nine years (2013–2021). Our main variable of interest is the indicator of disadvantaged community status (DAC), based on the U.S. Council on Environmental Quality's Climate and Environmental Justice dataset[49], which aligns with other studies using the same metric[81]. We also include a rich set of tract-level control variables $X$ (e.g., household income, population density, building age, the year residents moved into their current census tract, etc.), as well as county and year fixed effects to account for unobserved heterogeneity:

$$\begin{aligned} Count\ of\ Roof\ Renewals_{Tract,Year} = f(&\beta DAC_{Tract} + \gamma X_{Tract,Year} \\ &+ \delta_{County} + \phi_{Year} + \epsilon) \end{aligned}$$
$$(3)$$

In our main specification, we apply negative binomial count regression, which is particularly suited for overdispersed count data, because it explicitly models variance[82]. The model coefficients represent the expected change in the log count of the dependent variable $\hat{\mu}_{Tract}$ (the number of roof renewals within a tract) for a one-unit change in the predictor variable:

$$\hat{\mu}_{Tract} = e^{(\beta X_{Tract,Year} + \gamma_{County} + \varphi_{Year})} \quad (4)$$

Here, $\beta$ is a vector of regression coefficients for the independent variables, while $\gamma$ and $\varphi$ are spatial and temporal fixed effects, respectively. All independent variables $X$ are normalized to values between 0 and 1. To interpret these coefficients more tangibly, we exponentiated the coefficients for our plots, translating them into a factor by which the original count (i.e., the number of roof renewals) changes if the independent variables are at their maximum compared to their minimum.

The more parsimonious Poisson regression yields directionally similar coefficient estimates but fits the data less well (Supplementary Note 4). The Poisson regression is a special case of the negative binomial regression and assumes that the mean equals the variance[82]. We also refined our analysis to avoid conflating roof renewals with post-fire rebuilding by excluding buildings identified as damaged in the DINS database and all buildings within a 50-meter radius of destroyed buildings. Alternative analytical methods considering these damaged buildings did not substantially alter our regression outcomes, indicating that our findings are consistent and reliable.

## Identifying the effect of wildfire exposure on roof renewal decisions

Building on the building permit dataset from 2563 California census tracts, our empirical strategy aims to further identify the effect of wildfire exposure on roof renewal decisions. We employed a difference-in-differences strategy to compare changes in roof renewals for exposed tracts to unexposed tracts. To model the number of roof renewals on the census tract level we ran a negative binomial panel regression, which we specified as follows:

$$\begin{aligned} Roof\ Renewals_{T,Y} = f(&\beta * Exposure_{T,Y} * DAC_T * HO_{T,Y} + \gamma X_{T,Y} \\ &+ \delta_T + \phi_Y + \epsilon) \end{aligned}$$
$$(5)$$

The binary variable $Exposure_{T,Y}$ indicates that a census tract $T$ is defined as treated in year Y. To analyze treatment effect heterogeneity, we included DAC status and homeownership indicators, denoted as $HO_{it}$, along with their interaction effects with $Exposure_{T,Y}$ into the

regression analysis. $HO_{it}$ represents a binary variable that signifies whether the share of homeowner-occupied housing within a tract is below the 25th percentile. $X_{T,Y}$ is a vector of tract-level control variables (e.g., household income, population density, building age, the year residents moved into their current census tract, etc.). This binary approach simplifies interpretation, is less affected by outliers, and accommodates non-linear relationships better than continuous variables. Our analysis incorporates fixed effects, particularly census tract fixed effects $\delta_T$ to control for unobservable and time-invariant differences in roofing decisions across tracts, e.g., driven by varying planning departments (for the permits). Year fixed effects $\phi_Y$ account for temporal changes impacting roofing, such as macroeconomic shifts or legal changes.

We defined wildfire exposure as a binary variable $Exposure_{Tract,Year}$, which is equal to 1 if the population-weighted average distance to a large fire in the past three years was below 10 kilometers (6.21 miles). In accordance with the National Wildfire Coordination Group's classification of Class E or higher fires[83], we defined a fire to be large if it burned more than 300 acres (around 120 hectares). We hypothesized that homeowners closer to a fire would be more influenced, which also follows previous literature that finds highly localized treatment effects[84]. Given the longevity of roofs and the substantial size of the investment in combination with the process of finding a contractor and applying for a permit, we defined the treatment period to be three years after a fire. This assumption is also validated in our event-study type regression.

Our panel regression model is essentially a difference-in-differences design with staggered treatment timings. Thus, the identification of a treatment effect requires several assumptions such as a common trend between treatment and control groups. Following previous literature[85], we substantiate this assumption with an event-study type regression. A range of robustness tests aims at alleviating the secondary threat, which could arise from tract-specific, but time-variant variables, which we have not observed. Also, we implemented placebo treatments prior to a fire to test the possibility of fire affecting pre-treatment outcomes and reverse causality (Supplementary Note 10 for detailed discussion of threats to identification).

Our results remain both valid and statistically significant across a series of robustness tests. We integrated a rich set of controls and fixed effects and experimented with multiple model specifications. In some specifications, we added county-by-year fixed effects for annual legislative or political changes in counties. Our results are also robust to different specifications of the treatment variable (Supplementary Note 4 for detailed regression tables and robustness checks).

### Counterfactual simulations to quantify the potential of roofing initiatives for narrowing the equity gap in wildfire-induced destruction of residential buildings

We conducted counterfactual simulations to evaluate the impact of roof renewal rates on the future equity gap in wildfire-induced destruction of residential buildings in California. We simulated the expected number of destroyed residential buildings over the next 30 years for each census tract, considering different wildfire risk scenarios, the risk reduction potential of new roofs, and various roof replacement strategies.

To project future wildfire exposure and the average proportion of homes destroyed by wildfires for each census tract, we use the wildfire risk model from the First Street Foundation[75]. Their wildfire risk estimates are based on a 30 m spatial resolution model incorporating wildfire fuels, weather patterns, information on climate change, and structure vulnerabilities[56], and the model's results are also utilized by prior research to analyze future wildfire hazards[38,46,86]. The publicly available dataset classifies each building in a tract into 10 risk clusters. Based on historical data, we modeled roof renewal rates for DAC and non-DAC tracts and projected new roofs in disadvantaged and non-

disadvantaged tracts over 30 years, assuming a constant number of buildings. We tested three roof renewal strategies that differ in the order of re-roofing prioritization: (1) selecting buildings at random, giving rise to a uniform distribution across risk categories; (2) prioritizing highest-risk buildings within each tract; (3) focusing on the highest-risk buildings across all tracts. Each strategy was assessed at 10 incremental renewal rates (10% to 100%). High roof renewal rates (> 50%) were modeled to reduce wildfire exposure for adjacent lower-risk clusters of buildings within a tract. Our simulations incorporated three wildfire risk scenarios (minimum, maximum, and average risk) and three roof benefit scenarios (15%, 25%, and 35% reduction in destruction likelihood). For each combination of these scenarios and roof renewal strategies, we conducted 1000 simulation runs, totaling 270,000 samples per tract.

Destruction likelihood upon wildfire exposure was modeled using a beta distribution, with adjustments for new roofs and spillover effects in clusters of buildings with a high number of new roofs. We calculated the average destruction share across different scenarios and fitted a polynomial regression to estimate required roof renewal rates to close the equity gap between disadvantaged and non-disadvantaged communities. See Supplementary Note 11 for a detailed explanation of the counterfactual simulation and additional results.

### Analysis of Firewise communities

In our analysis, we focused on the over 2,000 Firewise communities across the United States that were active in 2022 (Supplementary Note 7). The dataset offers a snapshot of these communities, detailing when they were established, their resident population, and their cumulative investment over the years. To determine whether it falls within a disadvantaged community, we spatially match each Firewise community to the census tract that contains the point location that describes the community's location. We calculated the average annual investment by dividing the given total investment by the number of years the community has been active. Although this approach assumes a uniform distribution of investments across time, which is a simplification, it offers a valuable starting point for understanding the investment patterns of these communities. We suspect substantial survivorship bias within the data as it only contains information on the currently active communities. As a result, we have chosen to forego a formal regression analysis generalizing findings to a broader context or infer causality in favor of reporting descriptive statistics. Despite this limitation, our analysis illustrated existing patterns and trends of the Firewise communities.

### Reporting summary

Further information on research design is available in the Nature Portfolio Reporting Summary linked to this article.

## Data availability

All the data applied in our study were either acquired from public sources or were obtained via Freedom of Information Act and public records requests. CAL FIRE's Damage Inspection Database (DINS) is available at (https://www.fire.ca.gov/about/resources/california-public-records) (Request number R006742-013123). American Community Survey (ACS) 5-year estimates are available at the Census Bureau (https://data.census.gov/.). The Climate and Environmental Justice dataset is available at the U.S. Council on Environmental Quality (https://www.whitehouse.gov/environmentaljustice/justice40/). Building permits and assessor parcel maps were obtained from local authorities through open data portals or public requests. CAL FIRE's repository for fire perimeters is available through their GIS datacenter (https://frap.fire.ca.gov/mapping/gis-data/). Wildfire risk estimates from First Street are available for the public (https://firststreet.org/data-access/public-access/). Firewise communities data is available

through the NFPA (https://www.nfpa.org/Public-Education/Fire-causes-and-risks/Wildfire/Firewise-USA). The data for replicating the analyses presented in this study is deposited in the Open Science Framework under accession code https://osf.io/f8g94/?view_only=3f12a54e48a2442c876810fd1ad73156.

## Code availability

The code for replicating the analyses presented in this study is deposited in the Open Science Framework under accession code https://osf.io/f8g94/?view_only=3f12a54e48a2442c876810fd1ad73156.

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

## Acknowledgements

This work was funded in part by the European Union under the HORIZON program "Climate-Resilient Development Pathways in Metropolitan Regions of Europe (CARMINE)" (award number 101137851) (M.W. and D.N.). We thank the employees at city and county offices throughout

California for their assistance in acquiring the building permit and assessor data that was applied in this study.

## Author contributions

S.R., M.W., C.Z., and D.N. conceptualized and designed the research; S.R. performed the research and analyzed data; S.R., M.W., and C.Z. wrote the initial paper draft; all authors reviewed and edited the paper. D.N. provided funding acquisition support for the research.

## Funding

## Competing interests

The authors declare no competing interests.
