## [Transparent Peer Review file · Nature Communications]

Roof renewal disparities widen the equity gap in residential wildfire protection

Corresponding Author: Mr Sebastian Reining

Version 0:

Reviewer comments:

Reviewer #1

(Remarks to the Author)

Comments for Author

Overview:

This paper exhibits clear grounding in the existing literature and strength in the adopted methodology. Most areas that tend to influence, or might influence, the adoption or otherwise of roof hardening for homes in wildfire risk areas have been addressed to some depth. However, there are a few minor additions that might assist the non-US reader; likewise, those readers with a poorer handle on statistical analysis. Indeed, the greatest strength of this paper, its methodology, is also the area of greatest concern – not for lack of statistical proof, but in its slight excesses: to the extent that some highly useful and relevant descriptive material has been relegated to the supplementary file. These points aside, the paper is well written and offers significant guidance for federal and state governmental action both in the US and internationally, albeit in differing political, regulatory, built and natural environmental contexts.

Key results

Although the key results are, from one perspective, rather basic – i.e. that more non-DAC homes have roof renewals both pre and post wildfire – it is the deep research into the ‘why’ that is of significance here. Some might suggest that the ‘why’ is irrelevant, and that governments should just act to reduce the gap. However, governments act on figures, and the authors offer a wealth of data to back both conclusions and policy guidance. In addition, the data demonstrates just where funding should go specifically, i.e. the most at-risk DAC properties across all census tracts such that renewal rates surpass that of non-DAC buildings over the next 30 years. Though here to, some clarification is recommended and suggestions for doing so are offered later in this review.

Validity

Material is clearly statistically valid and, given the wealth of data and methodological description, highly replicable. Coming from a more qualitative perspective, it is also highly trustworthy given the background material surrounding the ‘numbers’. This is significant in grounding the chosen material contextually, giving cause to the data choices the authors have made. The authors make clear the California centric nature of their work, and the other logical limitations around vegetation types and management, varying wildfire intensities and housing contexts. However, as they suggest, the study provides further impetus for governments in other localities, indeed other countries, to engage proactively with the compounding issues of climate change.

This is not to say that that improvements in the contextual discussion supporting validity can’t be made, for suggestions are offered later in this review: these suggestions focus on a broadened interpretation of causation and an expanded discussion on the need for roof hardening deeper into communities (indeed cities) fringing bushlands and forests.

Significance

Is the work significant? As stated in the overview, the findings are not particularly startling, a basic socio-economic overview of the context offers the basis of much that is written here. But as also stated in the overview, it is the depth of data and analysis that makes the work important. It gives cause and direction to governmental action. It is data that is hard to refute. Despite the unstated philosophical or theoretical perspective of the authors, it is clear that they are offering a calm, statistical,

analysis that coldly states replicable facts. They have pointed to well known problem and shown cause for action, the manner of that and action, and how such action is to the security of the many, rather than the privileged few. In so doing, the paper provides crucial understandings for policymakers working towards mitigating the risks of wildfires while also promoting social equality. Thus, the study highlights a direct correlation between socioeconomic status and wildfire risk, underscoring the importance of targeted interventions.

Data and Methodology

The data capture in this work is broad and extremely well documented. Access to the raw data is also made easily accessible for those who would seek it, either for purposes of proving the findings, or for broadening the inquiry into other parts of home structures and or the surrounding context (built, natural, or Anthro-intervened [human controlled/influenced forests, farmlands, parklands and the like]). The author's data combinations (e.g. building permits, wildfire exposure, DAC designations) provides a comprehensive interpretation of roof renewal patterns relative to wildfire vulnerability. Such multi-source approaches enhance the analysis whilst avoiding reliance on self-reported data or limited datasets. That stated, it is appreciated that building permit data was identified by the authors as being notably incomplete regarding actual renewal numbers, particularly between DAC and non-DAC communities, and that future research needs to explore alternative means of enhancing this area of data capture.

Given their chosen reporting approach, particularly within the supplementary file, there is a wealth of material here for others to follow. However, if anything, there may be too much of a good thing in the main document – with the methodology section dominating the text, perhaps to the exclusion of other relevant explanatory contextual material. Some of the data is also presented through graphs in the main text that tend to confound rather than inform the reader (e.g. fig 1).

The latter points aside, the background code has been made available which allows further evidence checking of how the statistical analysis was developed and conducted. There appear to be no significant flaws in these approaches, and whilst some percentages of difference or temporality pan out to be seemingly minor, such findings are backed by levels of statistical significance (p) that in the main are demonstrably strong (0.05 – 0.001).

Analytical approach

Whilst statistics are the staple of the author's analysis, they have chosen to weave a range of approaches around the available data. Indeed, the number of approaches and the depth of their application is rather daunting. Counterfactual analysis has also been used, an important factor in confirming the relevance and effectiveness of the proposed interventions. Whilst the authors have attempted a level of qualitative analysis, the suggestions offered later in this review seek enhanced contextual depictions that ground the analysis and prevent it from becoming too esoteric.

Suggested improvements

There are but a few improvements that we suggest will help clarify this work:

- A definition (very brief) of a census tract
- A ref is required for lines 114-117
- Fig. 1: This 'looks' odd. 'Y' axis offers the number of buildings destroyed/100k, the decile columns offer the fine-grained data of exactly the number of buildings destroyed listed at the top. This three-layered data set is perhaps not necessarily the best way to present that material as there is a random and confusing correlation between these 'actually destroyed' numbers and the Y axis scale
- This is a more a question regarding a possibly important missing data set relating to Line 164: who owns the DAC non owner-occupied homes... do we assume owners are residents in the non-DAC regions and higher decile earnings? Or could they be owned by others in the DAC? Is there a 'care' factor here? I.e. do the owners consider insurance cover sufficient for their interests (i.e. the rebuilding of the structure, or a financed exit from the census tract) and do not concern themselves with occupant and or content survival?
- Lines 324-329: No correlation is offered between roof hardening and a lowering of fire suppression costs (i.e. the communities still need protection through wildfire suppression actions). Without this, it must be assumed that roof hardening cost will be additional to the protection strategy costs. A cost-effectiveness analysis comparing roof renewals to other mitigation strategies would be valuable for convincing policymakers
- In giving credence to the statistics behind the findings and discussion, the methodology section has become significantly large. Coupled with the Supplementary data it is a little heavy handed. Suggest a reduction in the word count here, and those words be used to include some of the more relevant supplementary material. E.g. Sup' fig' 4 and spread of Firewise communities across DACs; Key points from Sup note 9, offering slightly extended discussion on roofing refit permits in DACs and why else this may be low (this reviewer has noted that in some US states, local government oversight, and federal government funding, is significantly lacking in DACs seen to have low voting impact); likewise some material drawn from Sup notes 10 and 11. All of which would support the proposed approach to DAC roof hardening completions across at risk properties in all census tracts
- A (very brief) statement (and maybe an arial depiction or scaled mapping) of destruction patterns in towns and cities, as well as rural communities, would add weight the authors proposed actions and define more fully what is meant by at risk properties. Such descriptive imagery should highlight the capacity for ember attack and fire generated winds to destroy homes deep into the suburbs without the fire front ever getting there. The authors should seek currently applicable California data to align with their aims, however Australian examples include: Bianchi, R. & Leonard, J. (2005) Investigation of Bushfire Attack Mechanisms Resulting in House Loss in the ACT Bushfire 2003, Bushfire CRC Report, Canberra; and Costin G. P. (2021) Bushfire: Retrofitting Rural and Urban Fringe Structures – Implications of Current Engineering Data, Energies, Basel. The latter article highlighting homes destroyed in the Canberra 2003 fires a full 2 km from the fire front in the suburb of Lyons.

Clarity and context

Aside from the notes above, the article is clear and concise in the main. There is no arbitrary or irrelevant content that needs to be removed. The introduction and discussion sections effectively situate the research within the broader context of wildfire risk mitigation and social equity concerns. References to relevant literature and real-world examples like the California Wildfire Mitigation Program enhanced clarity and the research value.

Data Sources: The authors clearly state that all data came from public sources or public record requests.

References

A random assessment of the references has proven them to be relevant, genuine and current. Likewise, those that were offered in the supplementary file.

(Remarks on code availability)

The codes were accessible and downloadable. There was information sufficient to understand their application and installation.

Reviewer #2

(Remarks to the Author)

(Remarks on code availability)

I have accessed the code but the application of the code is outside of my expertise.

Reviewer #3

(Remarks to the Author)

The authors found that 1) disproportionately more homes were destroyed by wildfire in lower income census tracts in California from 2013-2021, and 2) that disadvantaged communities (DAC) had a substantial roof-renewal gap as compared to non-DAC in California. These are two completely separate analyses, but they connect them by positing that the roof renewal disparity gap contributes to greater structure loss (however, they do not test this hypothesis directly, since they do not have building permit data for the counties where most structure loss occurred). They further suggest through modeling that this disparity gap will widen into the future without efforts to increase roof renewal in DACs.

This is an interesting sliver of the complex wildfire puzzle, addressed through an economics modeling lens. There has been substantial literature on both the role of structural elements (i.e., materials, configuration, age) in wildfire home loss and there is an emerging literature on social vulnerability to wildfire, and this paper explores one very narrow facet of overlap, albeit an important one, by asking whether more socially vulnerable populations (as defined here by income) are less likely to replace their roofs with a newer one, making it more likely for the home to burn in a wildfire. It is consistent with prior literature in finding that yes, more socially vulnerable populations are at greater risk of fire-induced loss. It further attempts to quantify to the potential loss and cost savings that could be realized through increasing roof renewals in DACs.

However, it makes some key assumptions that reduce the broader appeal and applicability of the work. This is frustrating, because the general consensus among fire scientists is that grant programs to fund roof replacement and home hardening are absolutely critical to mitigating wildfire structure loss, which the authors detail in the discussion. But the assumptions made about relationships between roof renewal rates based on a few years of building permits and how much that would alter structure loss outcomes are a stretch, particularly to a non-economist. The paper is very well-written and well organized, and addressing concerns about assumptions and overreach in the modeling could make this paper more accessible and applicable. I really would have found the modeling far more believable if the authors had actually empirically connected disparities in roof renewal rates directly to increased structure loss by acquiring building permit data for the counties where most structures have been lost to wildfires, but they were unable to do so (as a side note, such data are available from vendors far more readily and more complete than from individual county offices).

I have some concerns about the validity of the assumptions that underly the analysis and the interpretation of the results. The authors have gone to great lengths to ensure their modeling is robust and develop counterfactuals that support their analysis, but they have failed to account for a few important factors in their initial data processing and extrapolation of their results through modeling. These include limitations in assessing roof-renewal rates, the failure to separate manufactured homes (mobile homes) from single-family homes (i.e., stick-built construction), and the nature of event-driven structure loss in wildfires (i.e., episodic loss events). It is unclear to me whether addressing these problematic assumptions would alter the results but I raise them to acknowledge the skepticism that policymakers might raise on this topic.

The three key problematic assumptions made herein are as follows:

1) Roof-renewal rates were calculated based on a reported 1.1 million building permits, however, these permits were pulled from a subset of 12 counties, as detailed in Supplemental Note 4. I found this interesting, because the authors' supplemental

figure that shows the spatial representation of census tracts for which the building permits were used and wildfire history shows that relatively few census tracts had fire history, primarily in remote areas of Shasta, northeastern Sierra, and eastern Kern Counties. Counties where the vast majority of wildfire structure loss in California has occurred over the last decade (Butte, Sonoma, Napa, Lake, Ventura, Santa Barbara, and Plumas) were not included in the building permit analysis, reducing its linkage to destructive wildfires. While the basic analysis of differential roof-renewal rates between DAC and non-DAC tracts in the assessed counties is unrelated to fire, the subsequent analysis of that gap increasing post-fire relies on the subset of counties selected and interprets this increasing gap as resident responding to perceived fire threats. In reality, few of those census tracts saw destructive fire.

A secondary concern with the roof renewal analysis is the lack of any discussion about the age of the structure or material. The authors make a key assumption here: all older roofs are flammable and newer roofs are not. This simply isn't true in California, where the Spanish tile construction that is widely favored in many of the hottest and most flammable parts of California have lifespans many decades longer (and is already incredibly fire-resistant). The authors only assessed 9 years of building permit data for roof renewals, and for only a limited number of census tracts. Given the spatial variability in growth rates across California, this begs the question of whether the growth timing was accounted for. Were the roof renewal gaps a product of timing of growth and ages of structures (or even the solar panel incentives, since higher-income homeowners often combine roof replacement and solar installation) rather than responses to largely non-destructive wildfires nearby?

2) I wondered why the authors failed to account for mobile homes and other "temporary" housing in their interpretation and their extrapolation of potential benefits, as this housing type is widespread and poses substantial barriers to home hardening in DACs. Roof renewals apply primarily to single family (stick-built homes), but mobile homes make up a substantial proportion of structures lost in wildfire, in the 2018 Camp Fire in Butte County, mobile homes comprised approximately 20% of the residential structures destroyed. Such housing stock would lower the roof renewal rates artificially in DACs with high mobile home proportions.

3) Finally, modeling the 30-year rate of wildfire-induced destruction based entirely upon disparity in roof renewals seems a bit of an overreach, particularly when the large body of literature that the authors themselves cite throughout points to the substantial complexity involved in trying to predict structure loss from wildfires, a challenge so difficult that numerous large insurers are unable to do so accurately enough to continue offering home insurance in California. Structure loss is episodic – there are hundreds to thousands of wildfires each year in the western US alone, and only a small fraction destroy any structures at all, with only a few destroying the vast majority of homes. The authors note that the DINS dataset assessed here includes some 33,689 structures, but over half of these (>18,000) are from just one wildfire, the 2018 Camp Fire, and just 5 fires make up nearly 90% of the dataset. This is particularly important because it begs the question of the relative roles of the extreme fire conditions versus the roof age of the structure in leading to structure loss.

I am not an expert in the type of statistical modeling utilized by the authors. But the assumptions I outline above raise concerns about the utility of the model outcomes.

I also wondered why the authors chose to use data from the national-level Justice40 project on fire risk and identifying DACs when California has its own state-level assessment of vulnerability from CalEnviroScreen, which is specific to the state and utilizes the key variables relevant to the state to define disadvantaged communities. Critically, the state index is also a continuous variable, rather than the binary classification (DAC or not) made by the The White House. California also has a state fire risk product from CALFIRE that is created in partnership with numerous fire agencies and reflects the conditions that are most relevant to fire risk in the state. Perhaps the authors ultimately intend a national analysis and wanted a national product, but because California has seen more destructive wildfires than any other state, it has developed more refined products to meet state needs.

(Remarks on code availability)

Reviewer #4

(Remarks to the Author)

The authors use a well established methodology; U.S. government's measure for disadvantaged communities (DACs). This part of the paper is well developed.

The main issues, however, are on the fire side. Using the roofs as surrogates for overall hardening is one component that does not by itself capture:

1. the overall hardening of a property (parcel and structure).

The roof is but one attribute of structure. I refer the authors to the NIST Hazard Mitigation Methodology (HMM), which has identified over 40 ember hardening vulnerabilities and many fire (radiation/convection) vulnerabilities.

<https://www.nist.gov/el/fire-research-division-73300/wildland-urban-interface-fire-73305/hazard-mitigation-methodology-2>.

Structure ignition is a very local phenomenon and is the result of an imbalance between exposures (fire and embers) and structure hardening. This local conditions do not average out in a community and therefore large scale evaluations require very high resolution exposure analyses that go beyond the documentation of one attribute (eg off covering).

2. the baseline construction of the building to be hardened.

All buildings are not created the same. The age of the building will impact its overall construction (codes change/improve over time). In CA the introduction of the CA WUI Building Code (Chapter 7A) has had significant impact on the construction of buildings. Additionally, the type of building and building setting have very significant effect on exposures. A single family home on a 1/3 acre lot, constructed after Chapter 7A was introduced, cannot be compared to a manufactured home from 1975 where the construction requirements were very different

3. the expected and received exposures (fire and embers).

Understanding exposures and how they impact structure hardening and by extension losses is essential to understanding the impact of Wildland Urban Interface fires on our communities. While embers travel hundreds and thousands of feet the impact of the flames through radiation and convection is extremely local. A manufactured or mobile home 10 to 15 away from a similar residence will have little chance of survival quasi independently of the roof age. The relations between fire exposures and structure response are spatially and temporally very complex and relate not only to the building, but also the surroundings including housing density.

While there are equity disparities that significantly impact community survivability and resilience to wildland and Wildland Urban Interface fires, looking just at the roofs is not the technically sound way to highlight them from a fire protection engineering perspective.

Regrettably I cannot recommend this manuscript for publication.

(Remarks on code availability)

Version 1:

Reviewer comments:

Reviewer #1

(Remarks to the Author)

Thank you for the manner in which you both received the review commentary, and the care and attention in its incorporation within the final draft. I say final advisedly, for I have no further concerns with the work, and I believe the other reviewers shall be equally approving. Your manner of attending to detail is to be applauded.

(Remarks on code availability)

As per original review.

Reviewer #2

(Remarks to the Author)

(Remarks on code availability)

As per the previous review.

Reviewer #3

(Remarks to the Author)

The authors have done a substantial amount of additional analysis in addressing the concerns raised by two of the reviewers. I find the results to be much more robust now, and the interpretation is more appropriate to the actual analysis conducted. Just a few comments:

1. On Figure 2a, error bars are provided for the roof renewal rate, but only a single value is shown for the equity gap. Since error bars are assigned, it seems like it would have been relatively simple to show a range in the equity gap to reflect that uncertainty.

2. Line 513 indicates that income deciles were based on 2019 ACS data, but data section (Line 475) describes the ACS data used as the 5-year estimates. So it's not actually the 2019 data, it's the 2015-2019 ACS data and needs to be represented as such to be accurate. Since the 5 year ACS estimates also have Margins of Error associated with them, how is this error addressed in your analysis? Given that you have broken the tracts into deciles and the MoE might place a census tract in a different decile based on the error rate, it seems like the results presented in Figure 1a could be affected by this.

3. Throughout the paper, it seems that there are two buildings datasets used derived from DINS: only residential buildings, and all buildings impacted by wildfires. It was unclear at several points if the dataset being analyzed for a particular question was the former (residential only) or the latter (all buildings), likely due to a desire to simplify the text. However, this is

confusing for the reader, and I might recommend either making sure you use the correct full terminology for each usage throughout, including in the figures (residential buildings vs buildings) OR using two different terms: 'homes' for residential buildings (or similar), and 'buildings' for all buildings, but also adding the caveat that you are using this terminology (which differs from DINS) to simplify representation of the data and reduce confusion.

(Remarks on code availability)

Reviewer #4

(Remarks to the Author)

While some changes have been made the fundamental premises of the article have not changes and my critical concerns remain the same.

The two fundamental issues are:

1. There is statement that roofs are important (with three references) and the Mazard Mitigation Methodology is mentioned but there is acknowledgment that assessing one attribute provides no meaningful context for the overall condition of the structure, parcel and actual exposures. From a fire protection engineering perspective, you can harden the roof and lose the house 100 % of the time from any of a number of embers or flame vulnerabilities. No data is presented that shows that roof replacements are clearly correlated to all the other hazards present.
2. Hardening needs are tied to exposures. This is only vaguely referenced in the report. Exposures are divided into fire and ember exposures, and further subdivided into direct and indirect exposures. An ember can land on the roof and if the roof is susceptible to ignition from embers can ignite the roof. You can have a new roof, and the ember can land on your and your neighbor's non-regulated shed and that will catch the eaves on fire and burn your house down or break the window and burn the house.

While replacing old worn roofs add value, without a complete structure/parcel and exposure analysis at the individual house level to assess the entire hazard/vulnerability balance, the presented work does not adequately caption the fire protection principles that drive fire spread in the WUI.

(Remarks on code availability)

Response to Reviewers: “Roof renewal disparities widen the equity gap in residential wildfire protection”

Reviewer comments in bold | Response to Reviewers in plain text

We would like to thank all of the reviewers for their insightful and thoughtful feedback about our research. In response to reviewer comments, we have collected additional data, conducted new analyses, refined and expanded the main text, and we have updated all of the figures and findings to reflect new data and analyses. Based on the reviewers' suggestions, we now include building permit data for four additional counties in California with high levels of historical wildfire exposure: Ventura County, Santa Barbara County, Napa County, and Butte County. While our initial dataset covered only 4.3% of residential wildfire damage in California, the newly added counties increased our coverage to 53.5%, as per the CAL FIRE DINS database. Our main findings were robust to the inclusion of this additional data, and enabled us to generate some additional insights, particularly linking historical roof renewals to reduced wildfire damage. We have also included several additional control variables and supplemental data to strengthen our findings. In responding to reviewer feedback we believe our article is now substantially improved.

Reviewer #1 (Remarks to the author):

Overview:

This paper exhibits clear grounding in the existing literature and strength in the adopted methodology. Most areas that tend to influence, or might influence, the adoption or otherwise of roof hardening for homes in wildfire risk areas have been addressed to some depth. However, there are a few minor additions that might assist the non-US reader; likewise, those readers with a poorer handle on statistical analysis. Indeed, the greatest strength of this paper, its methodology, is also the area of greatest concern – not for lack of statistical proof, but in its slight excesses: to the extent that some highly useful and relevant descriptive material has been relegated to the supplementary file. These points aside, the paper is well written and offers significant guidance for federal and state governmental action both in the US and internationally, albeit in differing political, regulatory, built and natural environmental contexts.

Key results

Although the key results are, from one perspective, rather basic – i.e. that more non-DAC homes have roof renewals both pre and post wildfire – it is the deep research into the ‘why’ that is of significance here. Some might suggest that the ‘why’ is irrelevant, and that governments should just act to reduce the gap. However, governments act on figures, and the authors offer a wealth of data to back both conclusions and policy guidance. In

addition, the data demonstrates just where funding should go specifically, i.e. the most at-risk DAC properties across all census tracts such that renewal rates surpass that of non-DAC buildings over the next 30 years. Though here to, some clarification is recommended and suggestions for doing so are offered later in this review.

Validity

Material is clearly statistically valid and, given the wealth of data and methodological description, highly replicable. Coming from a more qualitative perspective, it is also highly trustworthy given the background material surrounding the ‘numbers’. This is significant in grounding the chosen material contextually, giving cause to the data choices the authors have made. The authors make clear the California centric nature of their work, and the other logical limitations around vegetation types and management, varying wildfire intensities and housing contexts. However, as they suggest, the study provides further impetus for governments in other localities, indeed other countries, to engage proactively with the compounding issues of climate change.

This is not to say that that improvements in the contextual discussion supporting validity can’t be made, for suggestions are offered later in this review: these suggestions focus on a broadened interpretation of causation and an expanded discussion on the need for roof hardening deeper into communities (indeed cities) fringing bushlands and forests.

Significance

Is the work significant? As stated in the overview, the findings are not particularly startling, a basic socio-economic overview of the context offers the basis of much that is written here. But as also stated in the overview, it is the depth of data and analysis that makes the work important. It gives cause and direction to governmental action. It is data that is hard to refute. Despite the unstated philosophical or theoretical perspective of the authors, it is clear that they are offering a calm, statistical, analysis that coldly states replicable facts. They have pointed to well known problem and shown cause for action, the manner of that and action, and how such action is to the security of the many, rather than the privileged few. In so doing, the paper provides crucial understandings for policymakers working towards mitigating the risks of wildfires while also promoting social equality. Thus, the study highlights a direct correlation between socioeconomic status and wildfire risk, underscoring the importance of targeted interventions.

Data and Methodology

The data capture in this work is broad and extremely well documented. Access to the raw data is also made easily accessible for those who would seek it, either for purposes of proving the findings, or for broadening the inquiry into other parts of home structures and or the surrounding context (built, natural, or Anthro-intervened [human controlled/influenced forests, farmlands, parklands and the like]). The author’s data combinations (e.g. building permits, wildfire exposure, DAC designations) provides a comprehensive interpretation of roof renewal patterns relative to wildfire vulnerability. Such multi-source approaches enhance the analysis whilst avoiding reliance on self-reported data or limited datasets. That stated, it is appreciated that building permit data

was identified by the authors as being notably incomplete regarding actual renewal numbers, particularly between DAC and non-DAC communities, and that future research needs to explore alternative means of enhancing this area of data capture.

Given their chosen reporting approach, particularly within the supplementary file, there is a wealth of material here for others to follow. However, if anything, there may be too much of a good thing in the main document – with the methodology section dominating the text, perhaps to the exclusion of other relevant explanatory contextual material. Some of the data is also presented through graphs in the main text that tend to confound rather than inform the reader (e.g. fig 1).

The latter points aside, the background code has been made available which allows further evidence checking of how the statistical analysis was developed and conducted. There appear to be no significant flaws in these approaches, and whilst some percentages of difference or temporality pan out to be seemingly minor, such findings are backed by levels of statistical significance (p) that in the main are demonstrably strong (0.05 – 0.001).

Analytical approach

Whilst statistics are the staple of the author's analysis, they have chosen to weave a range of approaches around the available data. Indeed, the number of approaches and the depth of their application is rather daunting. Counterfactual analysis has also been used, an important factor in confirming the relevance and effectiveness of the proposed interventions. Whilst the authors have attempted a level of qualitative analysis, the suggestions offered later in this review seek enhanced contextual depictions that ground the analysis and prevent it from becoming too esoteric.

We sincerely thank the reviewer for their detailed feedback about our research. In responding to reviewer feedback, we have worked to clarify and refine the presentation of our methods and findings to improve their accessibility for those less familiar with the analytical approaches we deploy. We also have included research questions (page 3, paragraph 1) and other additions to the text to provide more motivation and subject relevance information for audiences with less exposure to the topic area and context. See below for point-by-point responses to the reviewer's queries.

Suggested improvements

There are but a few improvements that we suggest will help clarify this work:

- **A definition (very brief) of a census tract**

Thank you for your comment. As per your suggestion, we have included a short definition of a census tract based on the explanation provided by the United States Census Bureau (<https://www2.census.gov/geo/pdfs/education/CensusTracts.pdf>) in our methods section (page 16, paragraph 1).

- **A ref is required for lines 114-117**

Thank you for your comment. Your comment was referring to “*This disparity is accentuated at a per-property level due to the greater number of buildings in higher-income communities*”. We initially based our statement on our own analysis of the *American Community Survey (5-year estimates)* where we found that on average there are 1,830 housing units in non-DAC tracts compared to 1,570 in DAC tracts. We have, however, now also included two additional references that further supports this statement:

- (1) Dwyer, R. E. Expanding Homes and Increasing Inequalities: U.S. Housing Development and the Residential Segregation of the Affluent. *Social Problems* **54**, 23–46 (2007).
- (2) Blake, K., Kellerson, R. & Simic, A. Measuring Overcrowding in Housing. *U.S. Department of Housing and Urban Development Office of Policy Development and Research* (2007).

- **Fig. 1: This ‘looks’ odd. ‘Y’ axis offers the number of buildings destroyed/100k, the decile columns offer the fine-grained data of exactly the number of buildings destroyed listed at the top. This three-layered data set is perhaps not necessarily the best way to present that material as there is a random and confusing correlation between these ‘actually destroyed’ numbers and the Y axis scale**

Thank you for your comment. We agree that this figure was overly complex. To improve the visual presentation of information in Figure 1 and highlight the most important message, we have removed the “actually destroyed” numbers that were previously displayed as text above the bars to facilitate easier cross-decile comparisons. Additionally, in response to your comment about this figure we have also reviewed all of the figures in the text and made some adjustments to improve the presentation of information when appropriate (see, for example Figures 2 and 4).

- **This is a more a question regarding a possibly important missing data set relating to Line 164: who owns the DAC non owner-occupied homes... do we assume owners are residents in the non-DAC regions and higher decile earnings? Or could they be owned by others in the DAC? Is there a 'care' factor here? I.e. do the owners consider insurance**

cover sufficient for their interests (i.e. the rebuilding of the structure, or a financed exit from the census tract) and do not concern themselves with occupant and or content survival?

Thank you for your comment. Unfortunately, we do not have information on the current residency location of the owners of rental housing units. However, we do control for the share of owner-occupied households within a census tract and we also split the treatment effect by ownership rates (Fig. 3). Our findings support the reviewer's line of reasoning: owner-occupied houses are more likely to receive roof renewals than rental properties. In response to the reviewer's comment, we have adjusted the manuscript and added a brief interpretation of the higher roof renewal rates in owner-occupied areas (page 6, paragraph 1).

Fire risk is negatively correlated with the number of roof renewals, potentially related to stricter regulations that require more expensive roof installations. In contrast, areas with higher income, more new building construction, and a higher share of owner-occupancy, are associated with more roof renewals. This aligns with survey-based literature suggesting that financial capacity and homeowner presence are important drivers of wildfire protective actions, suggesting that a “care factor” — the willingness and ability of homeowners to invest in protective measures — might play a role in the decision to renew a roof.

Additionally, we have now added insurance coverage as a control variable in our regression analysis (Fig. 2B), which we obtained from California's Insurance Commissioner. Specifically, we matched their *Community Service Statement (CSS)* dataset, which contains information on insurance contracts per ZIP code from 2009 to 2020, with our roof renewal dataset. We calculated the number of insurance contracts per housing unit within each tract and included this insurance coverage in our regression analysis. The results indicate that the effect of insurance coverage on roof renewal is insignificant and does not affect our main findings.

California Department of Insurance: Community Service Statement (CSS) dataset (2009 - 2020).

Accessed on July 5, 2024. <https://www.insurance.ca.gov/>

• Lines 324-329: No correlation is offered between roof hardening and a lowering of fire suppression costs (i.e. the communities still need protection through wildfire suppression actions). Without this, it must be assumed that roof hardening cost will be additional to the protection strategy costs. A cost-effectiveness analysis comparing roof renewals to other mitigation strategies would be valuable for convincing policymakers

Thank you for your comment. Unfortunately, a comparison with other mitigation strategies is challenging to conduct due to a lack of comprehensive data and could perhaps be a research project of its own (and something we have indeed considered as potential future research). Previous studies suggest that a combination of multiple wildfire protection measures, which includes roof hardening, is likely the most effective strategy (e.g., Hakes et al., 2017). One of

our objectives is to examine the wildfire protection benefits associated with roof hardening. With this objective in mind, we have addressed the reviewer's feedback in two ways:

- (1) **Analysis of roof renewal benefits:** We analyzed the risk reduction potential of roof renewals in terms of wildfire damage using additional data from counties severely affected by wildfires and modeled the wildfire-induced damage on residential homes as a function of previous roof renewal rates. As shown in Fig. 1B and C, there is a negative and statistically significant relationship, demonstrating a strong correlation between roof hardening and reduced fire damage. The direction of this relationship, in turn, suggests the potential for lower wildfire suppression costs (although we caution against any interpretation beyond an association).
- (2) **Cost-benefit comparison using counterfactual simulation:** We conducted a cost-benefit analysis using the results of our counterfactual simulation. Starting with the expected number of houses destroyed by wildfires over the next 30 years, we calculated the number of homes saved based on various roof renewal strategies and associated costs. Therefore, we used the average house value on the ZIP Code level from the Zillow database (<https://www.zillow.com/research/data/>) together with a crosswalk mapping to U.S. census tracts provided by the Department of Housing and Urban Development and calculated the potential savings from avoided wildfire-induced building destruction net of estimated program costs. While this approach is simplified and does not account for the timing of fires, roof renewals, or discount rates, it also likely underestimates the benefits, as reduced fire damage would also lower wildfire suppression costs. To achieve equity in terms of destruction within the high wildfire risk zones of California, our base scenario indicates that upgrading approximately 527,000 houses (a 52% roof renewal rate) in DACs is necessary. With an average cost of \$20,000 per roof and potential funding mechanisms covering 30% of costs, the annual financial investment amounts to \$105 million – if the effort is evenly distributed over the next 30 years. These investments could potentially save around 10,000 houses in the DACs, valued at \$5 billion, representing a net gain of \$2 billion for the economy (Supplementary Note 12). In addition, roof renewals would likely address other vulnerabilities of disadvantaged communities, e.g., by reducing energy burden and making buildings “solar-ready”. We have summarized our findings in the discussion and Supplementary Note 12.

We hope this addresses your concerns and provides further insights about our research approach.

Hakes, R. S. P., Caton, S. E., Gorham, D. J. & Gollner, M. J. (2017). A Review of Pathways for Building Fire Spread in the Wildland Urban Interface Part II: Response of Components and Systems and Mitigation Strategies in the United States. *Fire Technol.* **53**, 475–515. <https://doi.org/10.1007/s10694-016-0601-7>

• In giving credence to the statistics behind the findings and discussion, the methodology section has become significantly large. Coupled with the Supplementary data it is a little

heavy handed. Suggest a reduction in the word count here, and those words be used to include some of the more relevant supplementary material. E.g. Sup' fig' 4 and spread of Firewise communities across DACs; Key points from Sup note 9, offering slightly extended discussion on roofing refit permits in DACs and why else this may be low (this reviewer has noted that in some US states, local government oversight, and federal government funding, is significantly lacking in DACs seen to have low voting impact); likewise some material drawn from Sup notes 10 and 11. All of which would support the proposed approach to DAC roof hardening completions across at risk properties in all census tracts

Thank you for your comment. In response, we streamlined the methods section to focus on the most critical information. In doing so, we moved large parts of the methods section that included less-critical information to the supplement (see, for example, the shortened methodological description of the counterfactual simulation and the new Supplementary Note 11) and further summarized and consolidated information to reduce word count where appropriate. In turn, we also elevated some of the material previously in the supplement to the main text, for example the analysis of the effect of fire resistant roofs on wildfire-induced building destruction (see Fig. 1C).

• A (very brief) statement (and maybe an arial depiction or scaled mapping) of destruction patterns in towns and cities, as well as rural communities, would add weight the authors proposed actions and define more fully what is meant by at risk properties. Such descriptive imagery should highlight the capacity for ember attack and fire generated winds to destroy homes deep into the suburbs without the fire front ever getting there. The authors should seek currently applicable California data to align with their aims, however Australian examples include: Bianchi, R. & Leonard, J. (2005) Investigation of Bushfire Attack Mechanisms Resulting in House Loss in the ACT Bushfire 2003, Bushfire CRC Report, Canberra; and Costin G. P. (2021) Bushfire: Retrofitting Rural and Urban Fringe Structures – Implications of Current Engineering Data, Energies, Basel. The latter article highlighting homes destroyed in the Canberra 2003 fires a full 2 km from the fire front in the suburb of Lyons.

Thank you for your comment. As per your suggestion, we have adjusted the manuscript accordingly, added some sentences and the corresponding references on the destruction patterns of wildfires in the introduction (page 2, paragraph 2).

Residential preparatory actions are critical for improved community protection, because the spread of fire within communities during a wildfire event is strongly related to the specific combustibility conditions of structures and their surrounding environment (Cohen, 2000). The primary cause of home destruction in wildfires is flying embers (Quarles et al., 2010 and Syphard & Keeley, 2019). Structures located up to 700 meters, and in some cases even as far as 2,000 meters, from the fire front can be destroyed during a wildfire (Costin, 2021 and Kershaw, 2003), and roofs, with their large surface areas, are particularly vulnerable to ignition (Blanchi et al., 2006 and Hakes et al., 2017).

- Blanchi, R., Leonard, J. & Leicester, R. Lessons learnt from post-fire surveys at the urban interface in Australia. *For. Ecol. Manag. - For. ECOL MANAGE* 234, (2006).
- Cohen, J. D. Preventing Disaster: Home Ignitability in the Wildland-Urban Interface. *J. For.* 98, 15–21 (2000).
- Costin, G. P. Bushfire: Retrofitting Rural and Urban Fringe Structures—Implications of Current Engineering Data. *Energies* 14, 3526 (2021).
- Hakes, R. S. P., Caton, S. E., Gorham, D. J. & Gollner, M. J. A Review of Pathways for Building Fire Spread in the Wildland Urban Interface Part II: Response of Components and Systems and Mitigation Strategies in the United States. *Fire Technol.* 53, 475–515 (2017).
- Kershaw, T. *Geoscience-Australia / Canberra Bushfires Fieldwork*. (2003).
- Quarles, S. L., Valachovic, Y., Nakamura, G. M., Nader, G. A. & de Lasaux, M. J. Home Survival in Wildfire-Prone Areas: Building Materials and Design Considerations. (University of California, Agriculture and Natural Resources, 2010). doi:10.3733/ucanr.8393.
- Syphard, A. & Keeley, J. Factors Associated with Structure Loss in the 2013–2018 California Wildfires. *Fire* 2, 49 (2019).

Clarity and context

Aside from the notes above, the article is clear and concise in the main. There is no arbitrary or irrelevant content that needs to be removed. The introduction and discussion sections effectively situate the research within the broader context of wildfire risk mitigation and social equity concerns. References to relevant literature and real-world examples like the California Wildfire Mitigation Program enhanced clarity and the research value.

We thank the reviewer for their assessment of our article.

Data Sources: The authors clearly state that all data came from public sources or public record requests.

References

A random assessment of the references has proven them to be relevant, genuine and current. Likewise, those that were offered in the supplementary file.

We thank the reviewer for checking the relevance of our references. In response to reviewer feedback, we have included 15 new references.

Reviewer #1 (Remarks on code availability):

The codes were accessible and downloadable. There was information sufficient to understand their application and installation.

We greatly appreciate that the reviewer accessed our code and data, and are happy to hear that it was understandable in its current format and presentation.

Reviewer #2 (Remarks to the author):

We are very supportive of this process and found the feedback from the reviewer to be extremely helpful in refining and improving our manuscript.

Reviewer #2 (Remarks on code availability):

I have accessed the code but the application of the code is outside of my expertise.

We sincerely thank the reviewer for their helpful feedback and the shared work with reviewer #1.

Reviewer #3 (Remarks to the author):

The authors found that 1) disproportionately more homes were destroyed by wildfire in lower income census tracts in California from 2013-2021, and 2) that disadvantaged communities (DAC) had a substantial roof-renewal gap as compared to non-DAC in California. These are two completely separate analyses, but they connect them by positing that the roof renewal disparity gap contributes to greater structure loss (however, they do not test this hypothesis directly, since they do not have building permit data for the counties where most structure loss occurred). They further suggest through modeling that this disparity gap will widen into the future without efforts to increase roof renewal in DACs.

This is an interesting sliver of the complex wildfire puzzle, addressed through an economics modeling lens. There has been substantial literature on both the role of structural elements (i.e., materials, configuration, age) in wildfire home loss and there is an emerging literature on social vulnerability to wildfire, and this paper explores one very narrow facet of overlap, albeit an important one, by asking whether more socially vulnerable populations (as defined here by income) are less likely to replace their roofs with a newer one, making it more likely for the home to burn in a wildfire. It is consistent with prior literature in finding that yes, more socially vulnerable populations are at greater risk of fire-induced loss. It further attempts to quantify to the potential loss and cost savings that could be realized through increasing roof renewals in DACs.

However, it makes some key assumptions that reduce the broader appeal and applicability of the work. This is frustrating, because the general consensus among fire scientists is that grant programs to fund roof replacement and home hardening are absolutely critical to mitigating wildfire structure loss, which the authors detail in the discussion. But the assumptions made about relationships between roof renewal rates based on a few years of building permits and how much that would alter structure loss outcomes are a stretch, particularly to a non-economist. The paper is very well-written and well organized, and addressing concerns about assumptions and overreach in the modeling could make this paper more accessible and applicable. I really would have found the modeling far more believable if the authors had actually empirically connected disparities in roof renewal rates directly to increased structure loss by acquiring building permit data for the counties where most structures have been lost to wildfires, but they were unable to do so (as a side note, such data are available from vendors far more readily and more complete than from individual county offices).

We sincerely thank the reviewer for their detailed comments about our research. In response to this feedback, we have made significant efforts to strengthen our manuscript by acquiring additional building permit data. We re-contacted all proposed county planning departments and engaged with the leading data vendor for building permit data in California. Unfortunately, despite our negotiations, the vendor's quoted price was prohibitively expensive. Additionally,

some heavily affected counties, such as Sonoma, established their databases in 2016, with complete data only available from 2020 onwards. However, we successfully obtained comprehensive building permit data through public records requests from Ventura County, Santa Barbara County, Napa County and Butte County. This substantially increased our dataset's coverage of residential wildfire damage from 4.3% to 53.5% in California, which in turn allowed us to expand our initial analyses with more observations and create new analyses, connecting the historical roof renewal rates to the structure loss dataset from CAL FIRE.

Please see below for our point-by-point responses to the reviewer's queries.

I have some concerns about the validity of the assumptions that underly the analysis and the interpretation of the results. The authors have gone to great lengths to ensure their modeling is robust and develop counterfactuals that support their analysis, but they have failed to account for a few important factors in their initial data processing and extrapolation of their results through modeling. These include limitations in assessing roof-renewal rates, the failure to separate manufactured homes (mobile homes) from single-family homes (i.e., stick-built construction), and the nature of event-driven structure loss in wildfires (i.e., episodic loss events). It is unclear to me whether addressing these problematic assumptions would alter the results but I raise them to acknowledge the skepticism that policymakers might raise on this topic.

The three key problematic assumptions made herein are as follows:

1) Roof-renewal rates were calculated based on a reported 1.1 million building permits, however, these permits were pulled from a subset of 12 counties, as detailed in Supplemental Note 4. I found this interesting, because the authors' supplemental figure that shows the spatial representation of census tracts for which the building permits were used and wildfire history shows that relatively few census tracts had fire history, primarily in remote areas of Shasta, northeastern Sierra, and eastern Kern Counties. Counties where the vast majority of wildfire structure loss in California has occurred over the last decade (Butte, Sonoma, Napa, Lake, Ventura, Santa Barbara, and Plumas) were not included in the building permit analysis, reducing its linkage to destructive wildfires. While the basic analysis of differential roof-renewal rates between DAC and non-DAC tracts in the assessed counties is unrelated to fire, the subsequent analysis of that gap increasing post-fire relies on the subset of counties selected and interprets this increasing gap as resident responding to perceived fire threats. In reality, few of those census tracts saw destructive fire.

Thank you for your comment. We have made substantial efforts to expand our dataset and improve coverage of census tracts, especially those recommended and which were historically affected by wildfire structure loss. We initially engaged in negotiations with a data vendor, but the quoted price was prohibitively expensive, even for singular counties. However, we re-contacted the seven counties proposed by the reviewer and successfully obtained data from

four additional counties which we have now included in our analysis. The table below contains an overview of our data acquisition actions and the responses from each county.

Table 1. New Data Acquisition Overview

County	Size (# Census Tracts)	Wildfire destroyed structures (in DINS)	Action / Response	Result
Butte	51	21,768	Public Records Request; provided all roofing permits (pre-filtered)	Included
Sonoma	100	8,066	Public Records Request; did provide roof renewal permits starting in 2020, as their database was only set up in 2016 and is incomplete	Not included
Napa	40	2,563	Public Records Request; provided all permits	Included
Lake	15	168	Public Records Request; did not provide roof renewal permits despite multiple attempts	Not included
Ventura	174	247	Public Records Request; provided all permits	Included
Santa Barbara	90	11	Public Records Request; provided all permits	Included
Plumas	7	1,375	Public Records Request to several offices; did not provide roof renewal permits despite multiple attempts	Not included

As outlined in other sections of this response to reviewers document, this new data has strengthened our research overall. Our initial findings (presented in the first version of the manuscript) were robust to the inclusion of these additional census tracts and importantly, the expanded dataset also enabled us to conduct a new analysis, connecting historic roof renewals to actual wildfire damages within a tract.

A secondary concern with the roof renewal analysis is the lack of any discussion about the age of the structure or material. The authors make a key assumption here: all older roofs are flammable and newer roofs are not. This simply isn't true in California, where the Spanish tile construction that is widely favored in many of the hottest and most flammable parts of California have lifespans many decades longer (and is already

incredibly fire-resistant). The authors only assessed 9 years of building permit data for roof renewals, and for only a limited number of census tracts. Given the spatial variability in growth rates across California, this begs the question of whether the growth timing was accounted for. Were the roof renewal gaps a product of timing of growth and ages of structures (or even the solar panel incentives, since higher-income homeowners often combine roof replacement and solar installation) rather than responses to largely non-destructive wildfires nearby?

Thank you for your comment. We do assume that new roofs are on average less flammable and our "roof benefit" analysis – based on the DINS database (previously detailed in Supplementary Note 5 and now illustrated in Fig. 1C) – generally supports this assumption. Our findings indicate that older structures, particularly those with older roofs, are on average more vulnerable to wildfires. Additionally, fire-resistant roofs have a stronger protective effect for houses in older age categories, as detailed in Supplementary Analysis No. 13. Moreover, due to California regulations, all new roofs in fire-prone areas are required to be fire resistant. In response to your comment, we have made several modifications to our manuscript:

- (1) **Roof benefit analysis based on CAL FIRE DINS database:** To control for differences in building age, we have included indicator variables for three age groups (constructed after 2008, constructed before 1990, or “construction year not observed” - buildings constructed between 1990 and 2008 serve as the reference group). As hypothesized, we find strong age-related effects: buildings constructed after 2008 are significantly less likely to be destroyed, while those built before 1990 are more susceptible to damage than the reference group. Our main findings regarding the benefits of fire-resistant roofs are robust to the inclusion of controls for building age. Also, we have elevated this analysis to the main part of the manuscript (now Fig. 1B and 1C, Supplementary Note 5 for detailed regression outputs).
- (2) **New analysis of interaction effects of fire-resistant roofs and construction year:** We investigated the interaction between fire-resistant roofs and building age (Supplementary Note 13). Our findings indicate that new roofs are beneficial, especially for buildings constructed before 2008. We observe nearly no effect for newer buildings. This is likely due to the fact that newer buildings are constructed with fire-resistant roofs, as prescribed by building codes since 2008. The strongest interaction is observed when the building age is unknown; in these cases, having a fire-resistant roof significantly reduces the risk of destruction.
- (3) **Analysis of differences in roof renewal rates:** We have incorporated two age groups as the share of buildings at the census tract level, which were “built before 1990” and “built after 2010” (with the missing age group “built between 1990 - 2010” serving as the reference group) in our analysis of differences in roof renewal rates (Fig. 2B), based on data from the *American Community Survey (5-year estimates)*. We have added this point in the discussion section to highlight this limitation (page 14, paragraph 1). Our analysis shows that tracts with a higher proportion of older buildings tend to have more roof renewals, but the DAC indicator is still negative and significant.

- (4) **New analysis, connecting historic roof renewal rates to actual wildfire damages:** Based on our newly acquired data for census tracts with high wildfire exposure, we examined the association of historic roof renewal rates and wildfire induced residential housing destruction. Our analysis shows that areas with the highest roof renewal rate in the past 3 years had 19% fewer wildfire-damaged houses compared to areas with the lowest share ($p < 0.01$, Fig. 2C). This analysis covers census tracts exposed to wildfires from 2013 to 2021 and accounts for about half of the residential building damage caused by wildfires in California during this period. We control for wildfire risk as well as other observed covariates.
- (5) **Controlling for residential solar installations:** In response to the reviewer's feedback we have included the residential solar installations per tract in a given year as a control variable in our regression model of differences in roof renewals. Solar installation data at census tract level was provided by Berkeley Lab's "*Tracking the Sun*" team. The coefficient of residential solar installation count is not statistically significant at the common $p < 0.05$ threshold and our results are robust to the inclusion of this additional control variable. Also, we would like to highlight that we removed any roof renewal that co-occurred with the installation of a solar PV system from our roof renewal dataset - this was already reflected in our initial submission.

Even though we find on average that building age is an appropriate proxy for vulnerability to wildfire, we do, however, now acknowledge that there are certain roofing materials, such as Spanish Tile mentioned by the reviewer, which could afford protection for older buildings that we cannot observe directly in our data. We have included this caveat as a limitation in the discussion section and have softened the language about this assertion in other areas of the manuscript (see page 14, paragraph 1).

Berkeley Lab. Tracking the Sun | Electricity Markets and Policy Group. <https://emp.lbl.gov/tracking-the-sun>.

2) I wondered why the authors failed to account for mobile homes and other “temporary” housing in their interpretation and their extrapolation of potential benefits, as this housing type is widespread and poses substantial barriers to home hardening in DACs. Roof renewals apply primarily to single family (stick-built homes), but mobile homes make up a substantial proportion of structures lost in wildfire, in the 2018 Camp Fire in Butte County, mobile homes comprised approximately 20% of the residential structures destroyed. Such housing stock would lower the roof renewal rates artificially in DACs with high mobile home proportions.

Thank you for bringing this to our attention. You make an excellent point, and we have now included information on mobile homes in multiple analyses appearing throughout the manuscript:

- (1) **Roof Benefit Analysis based on CAL FIRE DINS Database:** We have now included an indicator for mobile homes, which is positive and significant, thus confirming that

the risk is elevated for this structure type. Notably, the coefficient for fire-resistant roofs is robust to the inclusion of this variable and our main findings do not change.

- (2) **Analysis of differences in roof renewal rates:** We have included the share of mobile homes per census tract based on data from the *American Community Survey (5-year estimates)* in the regression. While the coefficient is strongly negative and significant, it neither affects the significance nor the magnitude of the DAC indicator. Moreover, we have also created a new version of the bar chart visualization, comparing the roof renewal rates over the years between DAC and non-DAC tracts, where we have adjusted the number of buildings within a census tract by the number of mobile homes (see Supplementary Figure 14). While the absolute numbers change, the overall pattern of less roof renewals in DAC tracts is robust to the inclusion of mobile homes.

3) Finally, modeling the 30-year rate of wildfire-induced destruction based entirely upon disparity in roof renewals seems a bit of an overreach, particularly when the large body of literature that the authors themselves cite throughout points to the substantial complexity involved in trying to predict structure loss from wildfires, a challenge so difficult that numerous large insurers are unable to do so accurately enough to continue offering home insurance in California.

Thank you for your comment. We acknowledge that wildfire exposure modeling is extremely complex and that large teams at research institutes, government agencies, and insurance companies are dedicated to improving model accuracy. To best leverage some of this existing domain expertise and resources, for our counterfactual simulation we used wildfire exposure models from the First Street Foundation published in 2022, which has also been utilized in other recent academic research studies exploring wildfire impacts on communities (see for example Boomhower (2023), Auer & Hexamer (2022) or Melecio-Vázquez et al. (2023)). The output from this model is also included in the U.S. government's Climate and Economic Justice Tool, which supports the Biden administration's Justice40 initiative. The First Street Foundation model has a 30 meter output resolution and includes factors such as climate change projections, vegetation, building characteristics and potential fire movements.

The risk projections from the First Street Foundation model are freely available at the census tract level and are therefore optimally suited for our counterfactual analysis. We highlight that there is a disparity in wildfire risk between DAC and non-DAC census tracts, which may be influenced by different exposure but also different building qualities, which First Street Foundation included in their model. Our primary contribution lies in analyzing roof renewals based on building permits rather than creating highly accurate exposure models. By incorporating differences in roof renewal rates into our model, we demonstrate how this disparity can worsen the existing exposure gap. Our modeling of roof benefits assumes a fixed risk reduction per scenario and an additional spillover benefits if more than 50% of buildings in a given risk category have new roofs. In response to your feedback, we have sharpened our main text to clarify these points (page 11, paragraph 1).

We chose not to make changes to the modeling approach that produced the outputs of our counterfactual simulation because it already involves many assumptions, and we did not want to introduce additional complexity. However, we have made the presentation of these simulation findings less prominent in our paper and placed more emphasis on other findings from our main analysis (note, for example, that previous Figures 4 and 5 are now merged into a reduced Figure 4 and that we included additional insights into Figure 1). We believe the simulation serves best as a data-driven evaluation of potential benefits rather than an exact projection of future wildfire impacts. Its purpose is to assess the scale of roof renewals necessary to make a significant difference and thereby assist in policy-making. Our manuscript has been adapted to reflect this perspective.

Auer, M. R. & Hexamer, B. E. Income and Insurability as Factors in Wildfire Risk. *Forests* **13**, 1130 (2022).

Boomhower, J. Adapting to growing wildfire property risk. *Science* **382**, 638–641 (2023).

Melecio-Vázquez, D. *et al.* A Coupled Wildfire-Emission and Dispersion Framework for Probabilistic PM_{2.5} Estimation. *Fire* **6**, 220 (2023).

Structure loss is episodic – there are hundreds to thousands of wildfires each year in the western US alone, and only a small fraction destroy any structures at all, with only a few destroying the vast majority of homes. The authors note that the DINS dataset assessed here includes some 33,689 structures, but over half of these (>18,000) are from just one wildfire, the 2018 Camp Fire, and just 5 fires make up nearly 90% of the dataset. This is particularly important because it begs the question of the relative roles of the extreme fire conditions versus the roof age of the structure in leading to structure loss.

Thank you for your comment. First, we wanted to share more insights into the DINS dataset which has data coverage through February 1, 2023. It includes 91,884 structures, of which 57,201 were damaged or destroyed, with 33,689 residential houses destroyed between 2013 and 2021. The “Camp Fire” alone accounted for 15,192 of these residential houses (i.e., ~45% of all destroyed residential buildings in the dataset), making it the most destructive fire on record. For some of our analyses, we use data on both damaged and undamaged houses, while for others, such as the overview of damages per income group, we filter the dataset to include only destroyed residential structures. In response to the reviewer’s comment, we have made several updates to our manuscript:

- (1) **Robustness test for roof benefit analysis:** We sequentially excluded the five largest fires from the analysis of the risk reduction potential of fire-resistant roofs, corresponding to in total 75% of the residential damage. Our results are robust to omission of those fire events. These findings are presented in Supplementary Table 12, which is part of Supplementary Note 5.
- (2) **Analysis of structures destroyed by wildfire across income deciles:** We conducted a similar robustness test by sequentially excluding the largest fire events and analyzing their impact on the structures destroyed by income deciles. The analysis, detailed in

Supplementary Figure 15, shows some fluctuation, with large fires influencing the relationship. However, the overall trend of lower-income communities suffering more damage is robust to this alteration.

We have also included an additional comment into our discussion, highlighting the episodic nature of wildfire loss (page 14, paragraph 1).

I am not an expert in the type of statistical modeling utilized by the authors. But the assumptions I outline above raise concerns about the utility of the model outcomes.

By including 4 additional counties, which includes the Camp Fire in Butte County, we now cover 53.5% of residential structural damage in the state of California from 2013 - 2022, which we believe now substantially bolsters the utility and robustness of our modeling outcomes. We hope this has helped address the reviewers' concerns.

I also wondered why the authors chose to use data from the national-level Justice40 project on fire risk and identifying DACs when California has its own state-level assessment of vulnerability from CalEnviroScreen, which is specific to the state and utilizes the key variables relevant to the state to define disadvantaged communities. Critically, the state index is also a continuous variable, rather than the binary classification (DAC or not) made by the The White House. California also has a state fire risk product from CALFIRE that is created in partnership with numerous fire agencies and reflects the conditions that are most relevant to fire risk in the state. Perhaps the authors ultimately intend a national analysis and wanted a national product, but because California has seen more destructive wildfires than any other state, it has developed more refined products to meet state needs.

Thank you for your comment. We indeed were aiming for broader applicability with our research. After also considering the comments from other reviewers, we have decided to continue using the Justice40 measure of disadvantaged communities. The binary measure is straightforward to interpret, aligns well with other studies using the same metric (e.g., Wussow et al., 2024), and is directly linked to U.S. Federal government financial commitments.

However, we have taken the reviewer's feedback into account and investigated the connection with the CalEnviroScreen vulnerability screening. In Supplementary Note 14, we recreate some of our main analyses using the CalEnviroScreen measure. Our findings indicate that, similar to the Justice40 measure, there is a significant negative relationship between the CalEnviroScreen Score (CI Score) and roof renewal rates. This shows that the trend of fewer roof renewals in more vulnerable communities holds true across both measures. For a more in-depth analysis of the population groups most affected by wildfires, we have also created breakdowns by age, education, unemployment, and ethnicity quintiles (Supplementary Note 14, Fig. 12).

Wussow, M. et al. Exploring the potential of non-residential solar to tackle energy injustice. *Nat Energy* 9, 654–663 (2024).

Reviewer #4 (Remarks to the author):

The authors use a well established methodology; U.S. government’s measure for disadvantaged communities (DACs). This part of the paper is well developed.

The main issues , however, are on the fire side. Using the roofs as surrogates for overall hardening is one component that does not by itself capture:

1. the overall hardening of a property (parcel and structure).

The roof is but one attribute of structure. I refer the authors to the NIST Hazard Mitigation Methodology (HMM), which has identified over 40 ember hardening vulnerabilities and many fire (radiation/convection) vulnerabilities. <https://www.nist.gov/el/fire-research-division-73300/wildland-urban-interface-fire-73305/hazard-mitigation-methodology-2>.

Structure ignition is a very local phenomenon and is the result of an imbalance between exposures (fire and embers) and structure hardening. This local conditions do not average out in a community and therefor large scale evaluations require very high resolution exposure analyses that go beyond the documentation of one attribute (eg off covering).

Thank you for your comment. We agree with the reviewer’s comment that roofs are but one of the elements of community survivability, which is a complex process with many facets, a point that we have made throughout the manuscript, and now even further emphasize, also by referencing the proposed NIST Hazard Mitigation Methodology. See below for examples of relevant text excerpts from the introduction and discussion sections:

Page 1, paragraph 3:

Wildfire preparedness is a multifaceted and perpetual process that encompasses risk prevention, reduction and preparation for potential evacuations. The most critical components of wildfire risk preparation for homeowners involve home hardening, which is the utilization of fire-resistant construction materials and design modifications to make structures less susceptible to fire, and vegetation management. Traditionally, the literature exploring wildfire preparedness has relied on household surveys and case studies, which are unable to capture the full geographic and temporal scale of the issue, as household mitigation actions are difficult to observe at scale. To address this observed gap in community impacts, we focus on the adoption of a critical household wildfire protection action: fire-resistant roofing, which can be observed by analyzing building permits.

First, while our focus on roofing and “Firewise communities” addresses key elements of wildfire preparedness, it is by no means exhaustive; other preventative measures such as vegetation management also contribute to wildfire resilience

However, many fire protection actions are difficult to observe at scale and related research that investigates effects of broad portfolios of intervention measures typically relies on household surveys with limited geographical and temporal scope (e.g., Dupey & Smith (2018), Wolters et al. (2017) and Brenkert-Smith et al. (2012)). Therefore, we chose to focus on a large-scale analysis of roof renewal decisions of individual households, which enables us to investigate disparities between communities of different types. In addition, we examined the distribution of “Firewise communities” across the U.S. as a low-cost alternative to roof renewals. Our findings indicate that while Disadvantaged Community (DAC) census tracts significantly lag behind non-DAC tracts in the number of “Firewise communities,” once established, these communities are equally active (Supplementary Note 7).

We also agree with the reviewer’s comment that structure ignition is a local phenomenon that requires high resolution exposure analysis and we acknowledge that such analyses are very complex with large teams at research institutes, government agencies, and insurance companies dedicated to developing related models and improving their accuracy. Therefore, we did not attempt to create another wildfire model but utilized the risk assessments that are provided by the Fire Factor Wildfire Risk Model from First Street Foundation. This peer-reviewed model (Kearnes, 2022) predicts the risk of wildfire damages at a 30m resolution over a 30 year timeframe and considers factors such as building and vegetation characteristics, local climate change impacts, and fire fuels. Its risk assessments are included in the U.S. government’s Climate and Economic Justice Tool, and they inform the disadvantaged community assignment by the Justice40 initiative. Moreover, the results are publicly available at the census tract level, enabling reproducibility of related research, and they have been utilized in various studies (see, for example, Boomhower (2023), Auer & Hexamer (2022) or Melecio-Vázquez et al. (2023)).

Our counterfactual simulation also clearly shows what the reviewer mentioned: Wildfire protection does not average out. Our analysis and the comparison of different roof renewal strategies shows that it is better to focus investments in a community on the highest risk areas, compared to spreading the home hardening across multiple areas (Fig. 4C).

Still, we acknowledge the parsimony of our simulation, the objective of which was to explore the potential of roof renewals and home hardening. At its core our simulation approach relies on assumptions about the reduction in risk of wildfire-induced building destruction that is achieved through roof renewal. An advantage is that it generalizes to the evaluation of other wildfire protection measures given assumptions about their risk reduction potential.

Auer, M. R. & Hexamer, B. E. Income and Insurability as Factors in Wildfire Risk. *Forests* **13**, 1130 (2022).

Boomhower, J. Adapting to growing wildfire property risk. *Science* **382**, 638–641 (2023).

- Brenkert-Smith, H., Champ, P. A. & Flores, N. Trying Not to Get Burned: Understanding Homeowners' Wildfire Risk–Mitigation Behaviors. *Environ. Manage.* 50, 1139–1151 (2012).
- Dupez, L. N. & Smith, J. An Integrative Review of Empirical Research on Perceptions and Behaviors Related to Prescribed Burning and Wildfire in the United States. *Environ. Manage.* 61, (2018)
- Kearns, E. J. et al. The Construction of Probabilistic Wildfire Risk Estimates for Individual Real Estate Parcels for the Contiguous United States. *Fire* 5, 117 (2022).
- Melecio-Vázquez, D. *et al.* A Coupled Wildfire-Emission and Dispersion Framework for Probabilistic PM_{2.5} Estimation. *Fire* 6, 220 (2023).
- Wolters, E. A., Steel, B. S., Weston, D. & Brunson, M. Determinants of residential Firewise behaviors in Central Oregon. *Soc. Sci. J.* 54, 168–178 (2017).

2. the baseline construction of the building to be hardened.

All buildings are not created the same. The age of the building will impact its overall construction (codes change/improve over time). In CA the introduction of the CA WUI Building Code (Chapter 7A) has had significant impact on the construction of buildings. Additionally, the type of building and building setting have very significant effect on exposures. A single family home on a 1/3 acre lot, constructed after Chapter 7A was introduced, cannot be compared to a manufactured home from 1975 where the construction requirements were very different

Thank you for your comment. We are in agreement with the reviewer that California's building code updates led to substantially improved fire-resistance of newer homes with CA WUI Building Code (Chapter 7A) introducing particularly effective home hardening measures. In response, we updated our analyses by incorporating more granular building age ranges that correspond to major updates of California's building codes as well as data on mobile homes. While our main findings are robust to the inclusion of these additional control variables, our analysis confirms that older buildings and mobile homes are more likely to get damaged when exposed to wildfires. Note the following updates to our manuscript:

- (1) **Roof benefit analysis based on CAL FIRE DINS database:** We have included more age groups to control for differences in building age and also included an indicator for mobile homes. In line with some of the reviewer's reasoning, we find strong age-related effects: buildings constructed after 2008 are significantly less likely to be destroyed, while those built before 1990 are more susceptible to damage. Also, structures identified as mobile homes by the DINS database are more likely to be destroyed. Nonetheless, our main findings regarding the benefits of fire-resistant roofs remain robust. Also, we have elevated this analysis to the main part of the manuscript (now Fig. 1B and 1C, Supplementary Note 5 for detailed regression output)
- (2) **New analysis of interaction effects of fire-resistant roofs and construction year:** We investigated the interaction between fire-resistant roofs and building age (Supplementary Note 13). Our findings indicate that new roofs decrease the risk of wildfire damages and that these risk reduction benefits increase with building age. The effect is not statistically significant for buildings constructed after 2008, which had to comply with CA WUI Building Code (Chapter 7A). Roof renewals reduce fire damage

risk most strongly for buildings with unknown construction year, which we expect to be relatively older.

- (3) **Analysis of differences in roof renewal rates:** We have incorporated more age groups as well as the number of mobile homes in our analysis of differences in roof renewal rates (Fig. 2B). This analysis shows that tracts with a higher proportion of older buildings tend to have more roof renewals, while a higher proportion of mobile homes correlates with a lower roof renewal rate. Our main results are robust to the inclusion of these additional control variables and the coefficient of the disadvantaged community indicator is neither affected in its magnitude nor in its significance. The distribution of building ages and structure types were extracted from the American Community Survey, which provides estimates at census tract level. We agree with the reviewer that an analysis of characteristics on building level could be an interesting addition to our research and we reflect this future research opportunity in our discussion (page 14, paragraph 1).

3. the expected and received exposures (fire and embers).

Understanding exposures and how they impact structure hardening and by extension losses is essential to understanding the impact of Wildland Urban Interface fires on our communities. While embers travel hundreds and thousands of feet the impact of the flames through radiation and convection is extremely local. A manufactured or mobile home 10 to 15 away from a similar residence will have little chance of survival quasi independently of the roof age. The relations between fire exposures and structure response are spatially and temporally very complex and relate not only to the building, but also the the surroundings including housing density.

While there are equity disparities that significantly impact community survivability and resilience to wildland and Wildland Urban Interface fires, looking just at the roofs is not the technically sound way to highlight them from a fire protection engineering perspective.

Regrettably I cannot recommend this manuscript for publication.

Thank you for your comment. Our primary aim is to highlight equity disparities in wildfire preparedness. Previous literature has often relied on small-scale case studies or surveys with limited participants, making it difficult to compare across communities. To address this challenge at a larger scale, we chose to focus on roof renewals as a preparedness action, because they are observable and quantifiable. Moreover, there is substantial research indicating that fire-resistant roofs are extremely important for wildfire mitigation – in combination with the building codes which the reviewer cited as well, we can therefore assume that roof renewals in fire-prone areas are also an investment in home hardening. However, we do not argue that roof renewals are the only important measure; we have clearly stated throughout our manuscript that other measures are also crucial (see our previous comments on page 18 and 19).

Also, modeling wildfire exposure several years into the future is indeed very complex. We present our simulation as a data-driven evaluation of potential benefits rather than an exact projection of future wildfire impacts. Its primary purpose is to assess the scale of roof renewals necessary to make a significant difference and thereby assist in policy-making. To address the complexity involved in modeling wildfires over such a long time horizon, we relied on the model developed by the First Street Foundation, which already considers many of the points raised by the reviewer (see our previous comments on page 19). We chose not to make any changes to the modeling behind our counterfactual simulation because it already involves numerous assumptions, and we did not want to introduce additional complexity.

To better reflect these points and to respond to reviewer's comments, we have made the following changes:

- (1) Added further controls, such as information on mobile homes, insurance coverage, residential solar adoption and building ages, to our analyses of roof benefits and roof renewal differences.
- (2) Conducted new supplementary robustness tests that highlight the risk reduction benefits of fire-resistant roofs across different building ages. Our regression also controls for various building-related factors, including neighbor density, structure types (e.g., single-family, mobile homes, multi-family home), and construction features (e.g., location of propane tanks, patios, exterior walls) (Supplementary Note 13).
- (3) Adapted our manuscript so it better reflects the limitations of our research and includes necessary caveats about the interpretation of our findings (see for example: page 13, paragraph 2; page 14 paragraph 1).

Response to Reviewers: “Roof renewal disparities widen the equity gap in residential wildfire protection”

Reviewer comments in bold | Response to Reviewers in plain text

We would like to thank all reviewers for their insightful and constructive feedback about our research. In response to the reviewers' comments, we have made several important revisions to improve the clarity and presentation of our research. Specifically, we have added further explanation to better support our methods and ensure transparency regarding the data used and included additional references where necessary. We have also revised several figures as suggested and have carefully reviewed the language used throughout the manuscript and made revisions where appropriate to prevent any misinterpretation regarding the risk reduction offered by new roofs. Furthermore, we now explicitly state in the discussion that roof renewals do not guarantee complete home protection and identify other important preventative factors, a point that we also make in the introduction. We greatly appreciate the reviewers' efforts and the valuable input they each provided, which has served to substantially improve our work.

Reviewer #1 (Remarks to the author):

Thank you for the manner in which you both received the review commentary, and the care and attention in its incorporation within the final draft. I say final advisedly, for I have no further concerns with the work, and I believe the other reviewers shall be equally approving. Your manner of attending to detail is to be applauded.

We sincerely thank the reviewer for their enthusiastic support of our research. The detailed feedback they provided in the previous round of review was invaluable in helping us elevate our manuscript.

Reviewer #2 (Remarks to the author):

We sincerely thank the reviewer for their helpful feedback and for their collaboration with reviewer #1. We would also like to reiterate our strong support for this initiative.

Reviewer #3 (Remarks to the author):

The authors have done a substantial amount of additional analysis in addressing the concerns raised by two of the reviewers. I find the results to be much more robust now, and the interpretation is more appropriate to the actual analysis conducted.

We sincerely thank the reviewer for these comments. We have addressed the additional feedback as outlined below.

Just a few comments:

1. On Figure 2a, error bars are provided for the roof renewal rate, but only a single value is shown for the equity gap. Since errors bars are assigned, it seems like it would have been relatively simple to show a range in the equity gap to reflect that uncertainty.

We thank the reviewer for this comment. In response to this feedback, we have added error bars around the estimates of the equity gap in Figure 2a, indicating 95% confidence intervals. We have also checked the other figures in the manuscript and did not find any additional instances where error bars needed to be included.

2. Line 513 indicates that income deciles were based on 2019 ACS data, but data section (Line 475) describes the ACS data used as the 5-year estimates. So it's not actually the 2019 data, it's the 2015-2019 ACS data and needs to be represented as such to be accurate. Since the 5 year ACS estimates also have Margins of Error associated with them, how is this error addressed in your analysis? Given that you have broken the tracts into deciles and the MoE might place a census tract in a different decile based on the error rate, it seems like the results presented in Figure 1a could be affected by this.

Thank you for your comment and for pointing this out. In line with the Justice40 initiative, we use the 2015-2019 ACS data and we have updated this accordingly in our manuscript. Previously, we indeed utilized only the point estimates of household median income from the 2015-2019 ACS data. In response to your feedback and to account for the uncertainty in classifying census tracts into income deciles based on ACS point estimates, we implemented a probabilistic simulation. This allowed us to test the robustness of our main finding: that the lowest three income deciles disproportionately suffer from wildfire-induced residential building destruction. We used the ACS point estimates and MoE to parameterize a normal distribution for the median income of each census tract. We drew the median incomes from these tract-specific distributions for each simulation run and classified census tracts into population-weighted income deciles. We repeated this procedure 10,000 times to analyze how the statistical variance affects our results. In the lowest three income deciles, the average number of destroyed buildings per 100,000 residential units is more than double that of the other deciles (476 buildings compared to 227 in the middle four and 228 in the highest three).

We therefore conclude that our main finding is robust to uncertainty introduced by the ACS' MoE. The newly created Supplementary Note 17 further details our simulation approach and results.

3. Throughout the paper, it seems that there are two buildings datasets used derived from DINS: only residential buildings, and all building impacted by wildfires. It was unclear at several points if the dataset being analyzed for a particular question was the former (residential only) or the latter (all buildings), likely due to a desire to simplify the text. However, this is confusing for the reader, and I might recommend either making sure you use the correct full terminology for each usage throughout, including in the figures (residential buildings vs buildings) OR using two different terms: 'homes' for residential buildings (or similar), and 'buildings' for all buildings, but also adding the caveat that you are using this terminology (which differs from DINS) to simplify representation of the data and reduce confusion.

We thank the reviewer for their comment. In response to this feedback, we have used the full terminology throughout the manuscript to indicate that we focus on roof renewals for residential buildings. For examples of where we made such changes in the text, see lines 66, 70, 290 (non-exhaustive), and the figure captions. We also added a sentence in our methodology to clarify our classification as “residential”.

Reviewer #4 (Remarks to the author):

While some changes have been made the fundamental premises of the article have not changes and my critical concerns remain the same.

The two fundamental issues are:

1. There is statement that roofs are important (with three references) and the Mazard Mitigation Methodology is mentioned but there is acknowledgment that assessing one attribute provides no meaningful context for the overall condition of the structure, parcel and actual exposures. From a fire protection engineering perspective, you can harden the roof and lose the house 100 % of the time from any of a number of embers or flame vulnerabilities. No data is presented that shows that roof replacements are clearly correlated to all the other hazards present.

We thank the reviewer for their comment. In response to this feedback and by the helpful guidance of the editor, we have adjusted the language in the entire manuscript to unambiguously reflect the on-the-ground reality that a new roof is no guarantee for a fire-safe house and is just one of several risk mitigation actions. In addition, we have added new statements in the introduction and the discussion to directly address the concerns that the reviewer has raised. See below for an example excerpt of this updated text in the revised manuscript:

Page 2, paragraph 1:

The most critical components of wildfire risk reduction for homeowners involve vegetation management, the reduction of fuels and home hardening, which is the utilization of fire-resistant construction materials and design modifications to make structures less susceptible to fire. For example, homeowners should use non-combustible siding, fences, and decking materials, install multi-pane windows with non-combustible shutters, and choose fire-resistant roofing.

2. Hardening needs are tied to exposures. This is only vaguely referenced in the report. Exposures are divided into fire and ember exposures, and further subdivided into direct and indirect exposures. An ember can land on the roof and if the roof is susceptible to ignition from embers can ignite the roof. You can have a new roof, and the ember can land on your and your neighbor's non-regulated shed and that will catch the eaves on fire and burn your house down or break the window and burn the house.

While replacing old worn roofs add value, without a complete structure/parcel and exposure analysis at the individual house level to assess the entire hazard/vulnerability balance, the presented work does not adequately caption the fire protection principles that drive fire spread in the WUI.

We again thank the reviewer for their comment. In response to this feedback and the helpful guidance of the editor, we have adjusted the introduction and the discussion, which now better reflect the different exposure types and the importance of parcel-level vulnerability assessments, whereby roof renewals are just one element. See below for examples of relevant text excerpts, which we have adapted or added:

Page 2, paragraph 2:

During wildfires, structures can be threatened by direct flame contact, radiant heat, and, most importantly, flying embers, which are the primary cause of residential building destruction.

Page 15, paragraph 1:

First, while our focus on roofing and “Firewise communities” addresses key elements of wildfire preparedness, it is by no means exhaustive; other home hardening measures such as wall, window, and door upgrades as well as vegetation management are also important for effective wildfire resilience. Homeowners and policy makers should note that comprehensive fire vulnerability assessments of structures are necessary, and roofs should be considered just one element among many actions required to ensure parcel security. Roof renewals alone do not guarantee home protection.

We would also like to note that we agree with the reviewer that a complete structure/parcel and exposure analysis at the individual household level likely represents an “ideal” way to conduct this research. However, this level of data granularity is currently unavailable at the state-wide scale that we pursue in our study. We have now mentioned this as future research in our discussion and the need for such high-resolution data collection efforts that extend beyond an event or single community analysis.